# DualCOIL: Offline Imitation Learning from Contrasting Demonstrations

**Huy Hoang** [1 2]  **Tien Mai** [1]  **Pradeep Varakantham** [1]  **Tanvi Verma** [3]

## Abstract

Offline imitation learning typically learns from expert and unlabeled demonstrations, yet often overlooks the valuable signal in explicitly undesirable behaviors. In this work, we study offline imitation learning from contrasting behaviors, where the dataset contains both expert and undesirable demonstrations along with an unlabeled set of demonstrations. We propose a novel formulation that optimizes a difference of KL divergences over the state-action visitation distributions of expert and undesirable (or bad) data. Although the resulting objective is a DC (Difference-of-Convex) program, we prove that it becomes *convex* when expert demonstrations outweigh undesirable demonstrations, enabling a practical and stable non-adversarial training objective. Our method avoids adversarial training and handles both positive and negative demonstrations in a unified framework. Extensive experiments on standard offline imitation learning benchmarks demonstrate that our approach consistently outperforms state-of-the-art baselines.

## 1. Introduction

Imitation learning (Garg et al., 2021; Kim et al., 2021; Li et al., 2023; Hoang et al., 2024a; Xu et al., 2022) offers a compelling alternative to Reinforcement Learning (RL) (Sutton & Barto, 2018; Puterman, 2014; Mnih et al., 2015) by enabling agents to learn directly from expert demonstrations without the need for explicit reward signals. This paradigm has been successfully applied in various domains, even with limited expert data, and is particularly effective in capturing complex human behaviors and preferences.

Imitation learning typically assumes access to high-quality expert demonstrations, which can be expensive and difficult to obtain (Ross et al., 2011; Torabi et al., 2018; Zhu et al., 2020). In practice, datasets often contain a mixture of expert and sub-optimal demonstrations. Recent advances in imitation learning have begun to address this more realistic setting, aiming to develop algorithms that can leverage informative signals from both expert and non-expert data (Brown et al., 2019; Myers et al., 2022; Hoang et al., 2024a).

In the offline setting, imitation learning methods typically assume the presence of a labeled expert dataset and an unlabeled dataset of mixed quality (which can contain expert, non-expert, and bad data), and further assume that the unlabeled demonstrations are not drastically different from expert behavior. This allows for framing the learning problem as mimicking both expert and unlabeled trajectories—albeit with different weights (Kim et al., 2021; 2022; Xu et al., 2022). However, in practice, unlabeled data may contain poor or undesirable demonstrations that the agent should explicitly avoid. For example, in autonomous driving, undesirable demonstrations may include unsafe lane changes or traffic violations, which should not be imitated under any circumstances. Another example can be found in healthcare applications, where undesirable demonstrations may correspond to incorrect diagnosis or unsafe treatment plans that could harm patients if imitated.

Unfortunately, existing imitation learning approaches are ill-equipped to deal with scenarios where both expert and undesirable demonstrations coexist (Wu et al., 2019; Zhang et al., 2021; Hoang et al., 2024a) whether in the main dataset or in the unlabeled dataset. It is important to note that learning by mimicking expert or mildly sup-optimal demonstrations is often tractable, as the corresponding objective—typically framed as divergence minimization—is convex (Kim et al., 2021; 2022). However, incorporating objectives that explicitly avoid bad (or undesirable) demonstrations can introduce non-convexities, making the optimization significantly more challenging. In this paper, we propose a unified framework that addresses the challenge of explicitly dealing with contrasting datasets (expert and undesirable) and an unlabeled dataset (expert, non-expert, undesirable demonstrations that are not labeled), aiming to bridge this gap in the current imitation learning literature.

---

[1]School of Computing and Information Systems, Singapore Management University, Singapore [2]Institute for Infocomm Research, Agency for Science, Technology and Research, Singapore [3]Institute of High Performance Computing, Agency for Science, Technology and Research, Singapore. Correspondence to: Huy Hoang <mh.hoang.2024@phdcs.smu.edu.sg>.

*Proceedings of the 43$^{rd}$ International Conference on Machine Learning*, Seoul, South Korea. PMLR 306, 2026. Copyright 2026 by the author(s).

Specifically, we focus on the setting of *offline imitation learning* (no interaction with the environment) where the dataset contains both *expert* and *undesirable* demonstrations[1] along with unlabeled demonstrations (contains unlabeled expert, non-expert and undesirable demonstrations). We make the following key contributions:

- We develop DUALCOIL (**Dual**-KL **CO**ntrastive **I**mitation **L**earning), a novel framework for learning from contrastive demonstrations, consisting of expert and undesirable trajectories. We formulate the learning objective as the difference of two KL divergences, yielding a *difference-of-convex* optimization problem. Although the learning objective is generally non-convex, we show that it becomes *convex* when the contribution of the expert demonstrations dominates that of the undesirable ones. This convexity is critical, as it enables us to reformulate the learning problem over the state-action visitation distribution as an more tractable unconstrained optimization via Lagrangian duality. Our objective stands in contrast to most existing distribution-matching imitation learning approaches, which typically rely solely on divergence minimization and naturally yield convex objectives. By introducing a divergence maximization term to account for undesirable behavior, we demonstrate that the overall objective *remains convex and manageable.*

- We further enhance the learning objective by proposing a surrogate objective that lower-bounds the original one, offering the advantage of a non-adversarial and convex optimization problem in the Q-function space. In addition, we introduce a novel Q-weighted behavior cloning (BC) approach, supported by theoretical guarantees, for efficient policy extraction.

- Extensive experiments on standard imitation learning benchmarks show that our method consistently outperforms existing approaches, both in conventional settings where datasets contain expert and unlabeled demonstrations, and in more realistic scenarios where explicitly undesirable demonstrations are included.

## 2. Related Works

**Imitation Learning.** Imitation learning trains agents to mimic expert behavior from demonstrations, with Behavioral Cloning (BC) serving as a foundational method by maximizing the likelihood of expert actions. However, BC often suffers from distributional shift (Ross et al., 2011). Recent work addresses this issue by leveraging the strong generalization capabilities of generative models (Zhao et al., 2023; Chi et al., 2023). Inspired by GANs (Goodfellow et al., 2014), methods like GAIL (Ho & Ermon, 2016) and AIRL (Fu et al., 2018) use a discriminator to align the learner's policy with the expert's, while SQIL (Reddy et al., 2019) simplifies reward assignment by distinguishing expert and non-expert behaviors. Although effective, these approaches typically require online interaction, which may be impractical in many real-world scenarios.

To address this, offline methods such as AlgaeDICE (Nachum et al., 2019) and ValueDICE (Kostrikov et al., 2020) employ Stationary Distribution Correction Estimation (DICE), though they often encounter stability issues. Building on ValueDICE, O-NAIL (Arenz & Neumann, 2020) avoids adversarial training, enabling stable offline imitation. More recently, several approaches have extended the DICE framework with stronger theoretical foundations and improved empirical performance (Lee et al., 2021; Mao et al., 2024). In parallel, IQ-Learn (Garg et al., 2021) has emerged as a unified framework for both online and offline imitation learning, inspiring a range of follow-up works (Al-Hafez et al., 2023; Hoang et al., 2024c). However, all these approaches rely on the presence of many expert demonstrations, which may not always be available.

**Offline imitation learning from suboptimal demonstrations:** Several approaches have been developed to tackle the challenges of offline imitation learning from suboptimal data, which is common in real-world scenarios. A notable direction involves preference-based methods, where algorithms infer reward functions by leveraging ranked or pairwise-compared trajectories to guide learning (Kim et al., 2023; Kang et al., 2023; Hejna & Sadigh, 2024). Recent works, such as SPRINQL (Hoang et al., 2024a), take advantage of demonstrations that exhibit varying levels of suboptimality, enabling the learner to better generalize beyond near-optimal behaviors. Another important line of research explores the use of unlabeled demonstrations in conjunction with a limited number of expert trajectories. Techniques like DemoDICE (Kim et al., 2021), SMODICE (Ma et al., 2022), and ReCOIL (Sikchi et al., 2024) apply Distribution Correction Estimation (DICE) (Sunehag et al., 2017; Lee et al., 2021; Mao et al., 2024) to re-weight trajectories and align the state or state-action distributions with those of the expert. In parallel, classifier-based methods, such as DWBC (Xu et al., 2022), ISW-BC (Li et al., 2023), and ILID (Yue et al., 2024), use discriminators to distinguish expert-like behaviors within mixed-quality data and assign them greater importance. Collectively, these strategies aim to enhance policy robustness and performance in offline settings where high-quality expert data is scarce or expensive to obtain. However, all of these approaches are primarily focused on imitating and are unable to avoid undesirable or

---

[1]In practice, while desirable demonstrations can be collected from expert decisions, undesirable ones can also be identified by experts (or even through fine-tuned LLMs (Mu et al., 2024))

bad demonstrations, which is crucial in domains such as self driving where there are many unsafe behaviors that would need to be avoided. There is prior work that focuses on learning explicitly from undesirable demonstrations (Jang et al., 2024; Hoang et al., 2024b), but these approaches cannot handle scenarios where both expert and undesirable datasets are available.

In this paper, we aim to optimize on the principle of "*Imitate the Good and Avoid the Bad*", which has recently gained attention in reference and safe RL (Abdolmaleki et al., 2025; Hoang et al., 2024a; Gong et al., 2025) and large language model training (Lu et al., 2025). We extend this idea to the offline imitation setting by proposing a novel and efficient method that learns from expert demonstrations while avoiding undesirable ones. To our knowledge, this is the first offline imitation learning approach to efficiently learn policies by jointly utilizing both expert and undesirable demonstrations.

## 3. Preliminaries

**Markov Decision Process (MDP).** We consider a MDP defined by the following tuple $\mathcal{M} = \langle S, A, r, P, \gamma, s_0 \rangle$, where $S$ denotes the set of states, $s_0$ represents the initial state set, $A$ is the set of actions, $r : S \times A \to \mathbb{R}$ defines the reward function for each state-action pair, and $P : S \times A \to S$ is the transition function, i.e., $P(s'|s, a)$ is the probability of reaching state $s' \in S$ when action $a \in A$ is made at state $s \in S$, and $\gamma$ is the discount factor. In reinforcement learning (RL), the aim is to find a policy that maximizes the expected long-term accumulated reward: $\max_\pi \left\{ \mathbb{E}_{(s,a) \sim d^\pi} [r(s, a)] \right\}$, where $d^\pi$ is the occupancy measure (or state-action visitation distribution) of policy $\pi$: $d^\pi(s, a) = (1 - \gamma)\pi(a|s) \sum_{t=1}^{\infty} \gamma^t P(s_t = s|\pi)$.

**Offline Imitation Leaning.** Recent imitation learning (IL) approaches have adopted a distribution-matching formulation, where the objective is to minimize the divergence between the occupancy measures (i.e., state-action visitation distributions) of the learning policy and the expert policy: $\min_{d^\pi} \left\{ D_f \left( d^\pi \parallel d^E \right) \right\}$, where $D_f$ denotes an $f$-divergence between the occupancy distributions $d^\pi$ (induced by the learning policy $\pi$) and $d^E$ (induced by the expert policy). In particular, when the Kullback–Leibler (KL) divergence is used, the learning objective becomes: $\min_{d^\pi} \ \mathbb{E}_{(s,a) \sim d^\pi} \left[ \log \left( \frac{d^\pi(s,a)}{d^E(s,a)} \right) \right]$. In the space of state-action visitation distributions ($d^\pi$), the training can be formulated as a convex constrained optimization problem. To enable efficient training, Lagrangian duality is typically employed to recast the problem into an unconstrained form (Lee et al., 2021; Kim et al., 2021).

**Offline IL with unlabeled data.** In offline imitation learning with unlabeled data, it is typically assumed that a limited set of expert demonstrations $\mathcal{B}^E$ is available, along with a larger set of unlabeled demonstrations $\mathcal{B}^{\text{Mix}}$. Distribution-matching approaches have been widely adopted to handle this setting. Prior methods often formulate the objective as a weighted sum of divergences between the learning policy and both expert and unlabeled data: $\min_{d^\pi} \left\{ D_f \left( d^\pi \parallel d^E \right) + \alpha D_f \left( d^\pi \parallel d^{\text{Mix}} \right) \right\}$, where $\alpha \geq 0$. Other approaches construct mixtures of occupancy distributions, such as $d^{\pi, \text{Mix}} = \alpha d^\pi + (1 - \alpha)d^{\text{Mix}}$ and $d^{E, \text{Mix}} = \alpha d^E + (1 - \alpha)d^{\text{Mix}}$, and minimize the divergence between $d^{\pi, \text{Mix}}$ and $d^{E, \text{Mix}}$ (Kim et al., 2021; 2022; Ma et al., 2022; Sikchi et al., 2024). In most existing approaches along this line of research, the convexity of the objective with respect to $d^\pi$ has been heavily leveraged to derive tractable learning objectives. However, when a divergence *maximization* term is introduced—as in our approach—this convexity may no longer hold, rendering many existing methods inapplicable.

## 4. DualCOIL: Offline Imitation Learning from Contrasting Behaviors

We begin by introducing a novel learning objective based on the difference between two KL divergences. Leveraging the convexity of this formulation, we derive a tractable and unconstrained optimization problem. Given that the resulting objective includes exponential terms that may lead to numerical instability, we enhance this by proposing a lower-bound approximation. This approximation enables us to reformulate the learning process as a more tractable, non-adversarial Q-learning objective, which remains convex in the space of Q-functions.

### 4.1. Dual KL-Based Formulation

Assume that we have access to three sets of demonstrations: good dataset $\mathcal{B}^G$ contains *good* or *expert* demonstrations, bad dataset $\mathcal{B}^B$ contains *bad* or *undesirable* demonstrations that the agent should avoid, and the unlabeled dataset $\mathcal{B}^{\text{Mix}}$ is a large set of unlabeled demonstrations used to support offline training. Here, we assume that $\mathcal{B}^B$ may contain low-reward or unsafe demonstrations that are undesirable to imitate, but *not necessarily behaviors that must be avoided at any costs*. Addressing such safety-critical cases would require alternative approaches based on hard-constrained or safe RL. We consider the realistic scenario where the identified datasets $\mathcal{B}^G$ and $\mathcal{B}^B$ are limited in size, while $\mathcal{B}^{\text{Mix}}$ is significantly larger—an assumption that aligns with typical settings in offline imitation learning from unlabeled demonstrations (Xu et al., 2022; Ma et al., 2022; Sikchi et al., 2024).

Let $d^\pi(s, a)$, $d^G(s, a)$, and $d^B(s, a)$ denote the state-action visitation distributions induced by the learned policy $\pi$, the good policy, and the bad policy, respectively. Following the DICE framework (Nachum et al., 2019; Kostrikov et al.,

2020), we propose to optimize the following objective:

$$\min_{d^\pi} \quad f(d^\pi) = D_{\mathrm{KL}}(d^\pi \,\|\, d^G) - \alpha \, D_{\mathrm{KL}}(d^\pi \,\|\, d^B), \quad (1)$$

where $\alpha > 0$ is a tunable hyperparameter. The goal of this objective is twofold: (1) to minimize the divergence between the learned policy and the good policy, and (2) to *maximize* the divergence from the bad policy, thereby avoiding undesirable behavior.

This formulation differs from all existing distribution matching approaches in the literature (Kim et al., 2021; Garg et al., 2021; Ma et al., 2022), which primarily focus on minimizing KL divergence—even when dealing with undesirable or unsafe demonstrations. By contrast, our approach introduces a principled mechanism to explicitly repel the learned policy from undesirable behavior while still aligning it with good data.

While the presence of a KL divergence maximization term in the objective may raise concerns about the convexity of the training problem, we observe that the objective in (1) takes the form of a difference between two convex functions. This is, in general, not convex and can be challenging to optimize. Fortunately, we show that under a mild condition, the overall objective remains convex. Specifically, if the weight on the bad policy divergence term is smaller than that on the good policy (i.e., $\alpha < 1$), then the objective becomes convex in $d^\pi$.

**Proposition 4.1.** *If $\alpha \leq 1$, then the objective function $f(d^\pi) = D_{KL}(d^\pi \,\|\, d^G) - \alpha \, D_{KL}(d^\pi \,\|\, d^B)$ is convex in $d^\pi$.*

Convexity is essential in most distribution-matching based frameworks, as it enables the use of Lagrangian duality to construct well-behaved and tractable training objectives. Our goal is to develop a Q-learning method that recovers a policy minimizing the objective in (1). To this end, we formulate the problem as the following constrained optimization:

$$\min_{d,\pi} \quad f(d, \pi) = D_{\mathrm{KL}}(d \,\|\, d^G) - \alpha \, D_{\mathrm{KL}}(d \,\|\, d^B) \quad (2)$$
$$\text{s.t.} \quad d(s, a) = (1 - \gamma)p_0(s)\pi(a \mid s)$$
$$+ \gamma\pi(a \mid s) \sum_{s', a'} d(s', a')T(s \mid s', a'), \; \forall s, a$$

where $d(s, a)$ is the state-action visitation distribution, and $T$ is the environment transition function. Let $\mathcal{B}^U = \mathcal{B}^G \cup \mathcal{B}^{\text{MIX}}$ denote the union dataset, and let $d^U$ be the state-action visitation distribution derived from it. The following proposition gives an another formulation for the objective in (1):

**Proposition 4.2.** *The objective function in (2) can be written as: $f(d, \pi) = (1 - \alpha)D_{KL}(d\|d^U) - \mathbb{E}_{(s,a)\sim d}\left[\Psi(s, a)\right]$, where $\Psi(s, a) = \log \frac{d^G(s,a)}{d^U(s,a)} - \alpha \log \frac{d^B(s,a)}{d^U(s,a)}$.*

This formulation introduces a KL-based regularization centered on the reference distribution $d^U$, with $\Psi(s, a)$ acting as a correction term that incorporates information from the labeled good and bad demonstrations. The reformulated objective in Proposition 4.2 further confirms that the function $f(d, \pi)$ remains convex in $d$ when $\alpha \leq 1$. Here we note that, under the same condition $\alpha \leq 1$, convexity may not hold for other $f$-divergences (a detailed discussion is provided in the appendix).

The reformulated objective in Proposition 4.2 takes the conventional form of maximizing a long-term surrogate reward $\Psi(s, a)$, subtracted by a KL divergence between two occupancy measures. Hence, we can reformulate it into a practical Q-learning objective (the detailed derivation is given in Appendix).

$$\max_\pi \min_Q \left\{ (1 - \gamma)\, \mathbb{E}_{(s,a)\sim p_0, \pi}\left[Q(s, a)\right] \right.$$
$$\left. + (1 - \alpha)\mathbb{E}_{d^U}\left[ e^{\frac{\Psi(s,a) + \gamma\,\mathbb{E}[Q(s',a')] - Q(s,a)}{1 - \alpha}} \right] \right\} \quad (3)$$

To further enhance the efficiency of Q-learning, we adopt the well-known Maximum Entropy (MaxEnt) reinforcement learning framework by incorporating an entropy term into the training objective (Garg et al., 2021; Haarnoja et al., 2018). This leads to the following objective:

$$L(Q, \pi) = (1 - \gamma)\, \mathbb{E}_{(s,a)\sim p_0, \pi}\left[ Q(s, a) - \beta \log \frac{\pi(a \mid s)}{\mu^U(a|s)} \right]$$
$$+ (1 - \alpha)\mathbb{E}_{d^U}\left[ e^{\frac{\Psi(s,a) + \gamma\,\mathbb{E}\left[ Q(s',a') - \beta \log \frac{\pi(a'|s')}{\mu^U(a'|s')} \right] - Q(s,a)}{1 - \alpha}} \right].$$

where $\mu^U(a|s)$ is the behavior policy representing the union dataset $\mathcal{B}^U$. We now define the soft value function and the soft Bellman operator as follows:

$$V_Q^\pi(s) = \mathbb{E}_{a\sim\pi(\cdot|s)}\left[ Q(s, a) - \beta \log \frac{\pi(a \mid s)}{\mu^U(a|s)} \right],$$
$$\mathcal{T}^\pi[Q](s, a) = Q(s, a) - \gamma\, \mathbb{E}_{s'\sim\mathcal{T}(\cdot|s,a)}\left[ V_Q^\pi(s') \right].$$

Using these definitions, the training objective can be rewritten as:

$$L(Q, \pi) = (1 - \gamma)\, \mathbb{E}_{s\sim p_0}\left[ V_Q^\pi(s) \right]$$
$$+ (1 - \alpha)\mathbb{E}_{d^U}\left[ \exp\left( \frac{\Psi(s,a) - \mathcal{T}^\pi[Q](s, a)}{1 - \alpha} \right) \right]. \quad (4)$$

Note that, when $\alpha = 1$, according to Proposition 4.2, the training objective reduces to a standard offline RL problem with reward function $\Psi(s, a)$: $\max_d \, \mathbb{E}_{(s,a)\sim d}\left[\Psi(s, a)\right] = \max \mathbb{E}\left[\sum_{t=0}^\infty \gamma^t \Psi(s_t, a_t)\right].$

### 4.2. Tractable Lower Bounded Objective

In this section, we propose an additional step to improve the stability and tractability of the learning objective introduced

above. We first observe that the exponential term in (4) may lead to instability during training. To address this issue, we propose to approximate the exponential using a linear lower bound, which not only improves stability but also preserves a similar optimization objective.

**Proposition 4.3.** *Let the surrogate objective be defined as:*

$$\widetilde{L}(Q, \pi) = (1 - \gamma) \, \mathbb{E}_{s \sim p_0} \left[ V_Q^\pi(s) \right]$$
$$- \mathbb{E}_{d^U} \left[ \delta(s, a) \mathcal{T}^\pi[Q](s, a) \right] + (1 - \alpha) \mathbb{E}_{d^U} \left[ \delta(s, a) \right]. \quad (5)$$

*where* $\delta(s, a) = \exp\left( \frac{\Psi(s,a)}{1-\alpha} \right)$. *Then* $\widetilde{L}(Q, \pi)$ *is a lower bound of* $L(Q, \pi)$, *with equality when* $\mathcal{T}^\pi[Q](s, a) = 0$ *for all* $(s, a)$.

The lower-bound approximation $\widetilde{L}(Q, \pi)$ offers several benefits. First, as a valid lower bound of $L(Q, \pi)$, maximizing $\widetilde{L}(Q, \pi)$ promotes the original objective. Second, its structure—linear in $Q$ and concave in $\pi$—leads to a simplified, non-adversarial training procedure (see Proposition 4.4). Finally, its optimization goals remain aligned with those of $L(Q, \pi)$, encouraging high expected soft value under the initial state distribution and consistency between the soft Bellman residual and the guidance signal $\Psi(s, a)$.

**Remark.** *The training objective in (5) generalizes the IQ-Learn objective (Garg et al., 2021) as a special case. In particular,* $\widetilde{L}(Q, \pi)$ *reduces exactly to the IQ-Learn objective when* $\alpha = 0$ *(i.e., the undesirable dataset is ignored) and* $\mathcal{B}^G \equiv \mathcal{B}^U$ *(i.e., the good dataset coincides with the union dataset). To see this, observe that when* $\alpha = 0$ *and* $d^G = d^U$, *the term* $\Psi(s, a)$ *becomes zero for all* $(s, a)$. *As a result, the surrogate objective simplifies to:* $\widetilde{L}(Q, \pi) = (1 - \gamma) \, \mathbb{E}_{s \sim p_0} \left[ V_Q^\pi(s) \right] - \mathbb{E}_{(s,a) \sim d^G} \left[ \mathcal{T}^\pi[Q](s, a) \right]$, *which is exactly the training objective proposed in IQ-Learn. Thus, our formulation can be viewed as a principled extension of IQ-Learn that explicitly accounts for and contrasts between good and bad behaviors.*

We now present several key properties of the training objective $\widetilde{L}(Q, \pi)$ that make it particularly convenient and tractable for use, as formalized in Proposition 4.4 below.

**Proposition 4.4.** $\widetilde{L}(Q, \pi)$ *is linear in* $Q$ *and concave in* $\pi$. *As a result, the max–min optimization* $\max_\pi \min_Q \widetilde{L}(Q, \pi)$ *can be equivalently reduced to the following non-adversarial problem:*

$$\min_Q \Big\{ \widetilde{L}(Q) = (1 - \gamma) \, \mathbb{E}_{s \sim p_0} \left[ V_Q(s) \right]$$
$$- \mathbb{E}_{(s,a) \sim d^U} \left[ \exp\left( \frac{\Psi(s,a)}{1-\alpha} \right) \mathcal{T}[Q](s, a) \right] \Big\},$$

*where the soft value function* $V_Q(s)$ *is defined as:* $V_Q(s) = \beta \log \left( \sum_a \mu^U(a|s) \exp(Q(s, a)/\beta) \right)$, *and the soft Bellman residual operator is given by:* $\mathcal{T}[Q](s, a) = Q(s, a) - \gamma V_Q(s)$. *Moreover* $\widetilde{L}(Q)$ *is convex in* $Q$.

## 5. Practical Algorithm

**Estimating Occupancy Ratios.** The training objective involves several ratios between state-action visitation distributions, which are not directly observable. These quantities can be estimated by solving corresponding discriminator problems. Specifically, to estimate the ratio $\frac{d^G(s,a)}{d^U(s,a)}$, we train a binary classifier $c^G : \mathcal{S} \times \mathcal{A} \to [0, 1]$ by solving the following standard logistic regression objective:

$$\max_{c^G} \{ \mathbb{E}_{d^G}[\log c^G(s, a)] + \mathbb{E}_{d^U}[\log(1 - c^G(s, a))] \}. \quad (6)$$

Let $c^{G*}(s, a)$ be optimal solution to this problem, then the ratio can be computed as: $\frac{d^G(s,a)}{d^U(s,a)} = \frac{c^{G*}(s,a)}{1 - c^{G*}(s,a)}$. Similar discriminators can be trained to estimate other ratios such as $\frac{d^B(s,a)}{d^U(s,a)}$.

**Implicit $V$-Update and Regularizers.** In the surrogate objective $\widetilde{L}(Q)$, the value function $V_Q$ is typically computed via a log-sum-exp over $Q$, which becomes intractable in large or continuous action spaces. To address this, we adopt Extreme Q-Learning (XQL) (Garg et al., 2023), which avoids the log-sum-exp by introducing an auxiliary optimization over $V$, jointly updated with $Q$. Specifically, $V$ is optimized using the *Extreme-V* objective: $J(V \mid Q) = \mathbb{E}_{(s,a) \sim d^U} \left[ e^{t(s,a)} - t(s, a) - 1 \right]$, where $t(s, a) = \frac{Q(s,a) - V(s)}{\beta}$. The main training objective with fixed $V$ is:

$$\widetilde{L}(Q \mid V) = (1 - \gamma) \, \mathbb{E}_{s \sim p_0} \left[ V(s) \right]$$
$$- \mathbb{E}_{d^U} \left[ \exp\left( \frac{\Psi(s,a)}{1-\alpha} \right) (Q(s, a) - \gamma \mathbb{E}_{s'}[V(s')]) \right]. \quad (7)$$

The overall optimization proceeds by alternating: (i) updating $Q$ via minimizing $\widetilde{L}(Q \mid V)$, and (ii) updating $V$ via minimizing $J(V \mid Q)$. Both sub-problems are convex, enabling efficient and stable training. To further enhance stability, we follow (Garg et al., 2021; 2023) and add a convex regularizer $\phi(\mathcal{T}[Q](s, a))$ to prevent reward divergence. We use the $\chi^2$-divergence, $\phi(t) = t^2/2$, a common choice in Q-learning.

**Policy Extraction.** Once the $Q$ and $V$ functions are obtained, a common approach for expert policy extraction is to apply advantage-weighted behavior cloning (AW-BC) (Kostrikov et al., 2021; Garg et al., 2023; Hejna & Sadigh, 2024; Sikchi et al., 2024):

$$\max_\pi \sum_{(s,a) \sim \mathcal{B}^U} \exp\left( \frac{1}{\beta} (Q(s, a) - V(s)) \right) \log \pi(a \mid s). \quad (8)$$

A key limitation of this formulation is that the value function $V(s)$ is only an approximate estimate from the Extreme-V objective, potentially introducing noise and bias into advantage computation and degrading policy quality. Inspired

by (Sikchi et al., 2025), to address this issue, we utilize a $Q$-only alternative that avoids reliance on $V(s)$. The following proposition shows that this $Q$-based objective can, in theory, recover the same optimal policy as the original advantage-weighted BC formulation.

**Proposition 5.1.** *The following $Q$-weighted behavior cloning (BC) objective yields the same optimal policy as the original advantage-weighted BC formulation in* (8)*:*

$$\max_{\pi} \sum_{(s,a) \sim \mathcal{B}^U} \exp\left(\frac{1}{\beta} Q(s,a)\right) \log \pi(a \mid s). \quad (9)$$

While the $Q$-weighted BC objective is theoretically equivalent to the advantage-weighted BC objective in terms of the optimal policy it recovers, it provides a simpler and more practical formulation. This simplification can lead to more stable and accurate optimization in practice. Our experimental results further demonstrate that the $Q$-weighted formulation consistently yields significantly better training outcomes compared to the advantage-weighted BC baseline. Bringing all components together, we present our DUALCOIL (**Dual**-KL **CO**ntrastive **I**mitation **L**earning) algorithm in Algorithm 1.

---

**Algorithm 1 DualCOIL**

---

**Require:** Datasets $\mathcal{B}^G$, $\mathcal{B}^B$, $\mathcal{B}^{\text{MIX}}$; training steps $N_\mu$, $N$;
   models: $c_{w_G}^G$, $c_{w_B}^B$, $\pi_\theta$, $Q_{w_q}$, $V_{w_v}$
1:  Assign $\mathcal{B}^U = \mathcal{B}^G \cup \mathcal{B}^{\text{MIX}}$
2:  *# Train discriminator $c_{w_G}^G$ and $c_{w_B}^B$*
3:  **for** $i = 1$ to $N_\mu$ **do**
4:     Update $(w_G, w_B)$ to minimize Objective 6.
5:  **end for**
6:  *# Train $Q_{w_q}$ and $V_{w_v}$, and policy $\pi_\theta$*
7:  **for** $i = 1$ to $N$ **do**
8:     Update $w_q$ to minimize $\widetilde{F}(Q_{w_q}|V_{w_v})$
9:     Update $w_v$ to minimize $J(V_{w_v}|Q_{w_q})$
10:    Update $\theta$ via QW-BC:
       $\max_{\pi} \left\{ \sum_{(s,a) \sim \mathcal{B}^U} e^{Q(s,a)/\beta} \log \pi(a|s) \right\}$
11: **end for**

---

## 6. Experiments

In this section, we present extensive experimental results to evaluate the effectiveness of our method, focusing on the following key questions: **(Q1)** Can DualCOIL effectively leverage both labeled good and bad data to outperform existing baselines? **(Q2)** How does the size of the bad dataset $\mathcal{D}^B$ influence the performance of DualCOIL? **(Q3)** Can existing offline RL methods exploit the reward function $\Psi(s, a)$ as effectively as DualCOIL? **(Q4)** DualCOIL relies on a critical parameter $\alpha$ to balance objectives derived from good and bad data—how sensitive is the performance to this parameter?

Furthermore, in the Appendix, we provide comprehensive additional analyses (e.g., comparison between the original exponential-form objective and its surrogate), omitted proofs, and further experimental results that validate our approach.

### 6.1. Experiment setting

**Environments and Dataset Generation.** We evaluate our method in the context of learning from the good dataset $\mathcal{B}^G$ and avoid the bad dataset $\mathcal{B}^B$ with a support from an additional unlabeled dataset $\mathcal{B}^{\text{MIX}}$. Our experiments span four MuJoCo locomotion tasks: CHEETAH, ANT, HOPPER, WALKER, as well as four hand manipulation tasks from Adroit: PEN, HAMMER, DOOR, RELOCATE, and one task from FrankaKitchen: KITCHEN—all sourced from the official D4RL benchmark (Fu et al., 2020). For each MuJoCo task from D4RL, we have three types of datasets: RANDOM, MEDIUM, and EXPERT. The good dataset $\mathcal{B}^G$ is constructed using a single trajectory from the EXPERT dataset. The bad dataset $\mathcal{B}^B$ consists of 10 trajectories selected from either the RANDOM or MEDIUM dataset. To construct the unlabeled dataset $\mathcal{B}^{\text{MIX}}$, we combine the entire RANDOM or MEDIUM dataset (i.e., the same source as $\mathcal{B}^B$) with 30 additional trajectories from the EXPERT dataset. This setup mirrors the challenging RANDOM+FEW-EXPERT and MEDIUM+FEW-EXPERT scenarios introduced in Re-COIL (Sikchi et al., 2024). These three datasets—$\mathcal{B}^G$, $\mathcal{B}^B$, and $\mathcal{B}^{\text{MIX}}$—form the foundation of our training pipeline. We use the same dataset construction strategy for Adroit and FrankaKitchen tasks, yielding 18 distinct dataset combinations. Please refer to the Appendix for detailed descriptions of all dataset combinations.

**Baselines.** We compare our method against several baselines. First, we evaluate two naive BC approaches: one that learns directly from the large unlabeled dataset $\mathcal{B}^{\text{MIX}}$ (BC-MIX), and one that learns solely from the good dataset $\mathcal{B}^G$ (BC-G). Next, we include comparisons with state-of-the-art methods designed to leverage both expert (or good) data $\mathcal{B}^G$ and unlabeled data $\mathcal{B}^{\text{MIX}}$, including SMODICE (Ma et al., 2022), ILID (Yue et al., 2024), and ReCOIL (Sikchi et al., 2024). We exclude DWBC (Xu et al., 2022) from this experiment since both DWBC and ILID use discriminator-based objectives, and ILID has been shown to outperform DWBC. In addition, based on our proposed objective in (5), we include a variant of our method that only learns from $\mathcal{B}^G$ and $\mathcal{B}^{\text{MIX}}$ (i.e., $\alpha = 0$), called as DualCOIL-G. For methods that incorporate support from bad data $\mathcal{B}^B$, we evaluate our approach against SafeDICE (Jang et al., 2024). Given the limited number of existing baselines that effectively utilize poor-quality data in offline imitation learning, we also propose a simple adaptation of DWBC, which is called as DWBC-GB to jointly learn from $\mathcal{B}^G$, $\mathcal{B}^B$, and $\mathcal{B}^{\text{MIX}}$. Detailed implementation of these baselines are provided in the

# DualCOIL: Offline Imitation Learning from Contrasting Demonstrations

| Task | unlabeled $\mathcal{B}^{\text{MIX}}$ | learning from $\mathcal{B}^G$ and $\mathcal{B}^{\text{MIX}}$ only | | | | | | learning with $\mathcal{B}^B$ | | | |
|---|---|---|---|---|---|---|---|---|---|---|---|
| | | BC-MIX | BC-G | SMODICE | ILID | ReCOIL | DualCOIL-G | SafeDICE | DWBC-GB | DualCOIL | Expert |
| CHEETAH | RANDOM+EXPERT | $2.3_{\pm0.0}$ | $-0.6_{\pm0.3}$ | $4.6_{\pm1.2}$ | $21.1_{\pm3.4}$ | $2.0_{\pm0.3}$ | $84.4_{\pm2.4}$ | $-0.0_{\pm0.0}$ | $2.8_{\pm0.5}$ | $\mathbf{86.7}_{\pm2.2}$ | 90.6 |
| | MEDIUM+EXPERT | $42.5_{\pm0.2}$ | $-0.6_{\pm0.3}$ | $42.4_{\pm1.6}$ | $40.3_{\pm7.0}$ | $42.5_{\pm0.3}$ | $48.6_{\pm2.0}$ | $37.7_{\pm0.1}$ | $5.6_{\pm1.9}$ | $\mathbf{77.6}_{\pm3.6}$ | 90.6 |
| ANT | RANDOM+EXPERT | $30.9_{\pm0.0}$ | $-7.2_{\pm4.6}$ | $4.6_{\pm9.7}$ | $71.8_{\pm8.7}$ | $56.2_{\pm5.0}$ | $100.6_{\pm9.9}$ | $-2.6_{\pm0.0}$ | $6.5_{\pm3.4}$ | $\mathbf{112.7}_{\pm5.8}$ | 117.5 |
| | MEDIUM+EXPERT | $91.2_{\pm0.8}$ | $-7.2_{\pm4.6}$ | $88.5_{\pm4.2}$ | $39.6_{\pm11.5}$ | $100.8_{\pm4.0}$ | $102.4_{\pm3.5}$ | $88.1_{\pm0.4}$ | $-4.3_{\pm2.4}$ | $\mathbf{107.4}_{\pm4.9}$ | 117.5 |
| HOPPER | RANDOM+EXPERT | $4.9_{\pm0.1}$ | $17.9_{\pm2.7}$ | $56.4_{\pm9.2}$ | $81.6_{\pm14.3}$ | $81.0_{\pm14.7}$ | $79.4_{\pm14.8}$ | $41.1_{\pm1.4}$ | $40.8_{\pm9.5}$ | $\mathbf{93.6}_{\pm9.2}$ | 109.6 |
| | MEDIUM+EXPERT | $52.2_{\pm0.6}$ | $17.9_{\pm2.7}$ | $53.0_{\pm1.7}$ | $87.9_{\pm5.3}$ | $46.1_{\pm8.3}$ | $70.6_{\pm8.0}$ | $55.8_{\pm1.7}$ | $21.6_{\pm4.0}$ | $\mathbf{103.7}_{\pm7.3}$ | 109.6 |
| WALKER | RANDOM+EXPERT | $1.5_{\pm0.0}$ | $3.8_{\pm1.5}$ | $106.6_{\pm0.7}$ | $100.1_{\pm4.4}$ | $29.8_{\pm14.9}$ | $97.5_{\pm10.7}$ | $23.0_{\pm0.8}$ | $17.4_{\pm7.5}$ | $\mathbf{107.4}_{\pm1.7}$ | 107.7 |
| | MEDIUM+EXPERT | $70.8_{\pm0.3}$ | $3.8_{\pm1.5}$ | $6.0_{\pm2.2}$ | $89.7_{\pm10.6}$ | $72.1_{\pm5.4}$ | $99.8_{\pm6.9}$ | $60.2_{\pm1.3}$ | $25.6_{\pm7.4}$ | $\mathbf{108.2}_{\pm0.4}$ | 107.7 |
| PEN | CLONED+EXPERT | $56.0_{\pm0.5}$ | $8.8_{\pm1.4}$ | $10.9_{\pm6.5}$ | $1.9_{\pm2.1}$ | $79.2_{\pm9.6}$ | $66.3_{\pm9.6}$ | $19.9_{\pm2.1}$ | $9.5_{\pm3.9}$ | $\mathbf{96.4}_{\pm8.7}$ | 107.0 |
| | HUMAN+EXPERT | $18.3_{\pm0.6}$ | $8.8_{\pm1.4}$ | $-2.5_{\pm0.2}$ | $5.1_{\pm2.1}$ | $99.9_{\pm8.5}$ | $95.5_{\pm8.8}$ | $21.8_{\pm2.5}$ | $6.5_{\pm2.4}$ | $\mathbf{101.5}_{\pm8.4}$ | 107.0 |
| HAMMER | CLONED+EXPERT | $0.4_{\pm0.4}$ | $1.4_{\pm0.3}$ | $0.8_{\pm0.4}$ | $0.4_{\pm0.6}$ | $3.4_{\pm2.1}$ | $66.5_{\pm11.8}$ | $0.0_{\pm0.1}$ | $2.8_{\pm2.5}$ | $\mathbf{74.3}_{\pm8.0}$ | 119.0 |
| | HUMAN+EXPERT | $12.8_{\pm3.3}$ | $1.4_{\pm0.3}$ | $1.9_{\pm2.1}$ | $1.2_{\pm1.4}$ | $113.2_{\pm5.5}$ | $113.2_{\pm7.2}$ | $0.6_{\pm0.4}$ | $3.4_{\pm1.9}$ | $\mathbf{120.0}_{\pm3.7}$ | 119.0 |
| DOOR | CLONED+EXPERT | $0.4_{\pm0.3}$ | $-0.1_{\pm0.0}$ | $-0.1_{\pm0.0}$ | $-0.1_{\pm0.1}$ | $19.3_{\pm7.5}$ | $92.6_{\pm5.1}$ | $-0.0_{\pm0.0}$ | $-0.1_{\pm0.0}$ | $\mathbf{102.4}_{\pm1.7}$ | 105.3 |
| | HUMAN+EXPERT | $4.0_{\pm1.2}$ | $-0.1_{\pm0.0}$ | $-0.1_{\pm0.3}$ | $0.2_{\pm0.7}$ | $100.3_{\pm2.9}$ | $104.7_{\pm0.7}$ | $0.9_{\pm0.4}$ | $1.1_{\pm0.5}$ | $\mathbf{105.0}_{\pm0.5}$ | 105.3 |
| RELOCATE | CLONED+EXPERT | $-0.1_{\pm0.0}$ | $-0.1_{\pm0.0}$ | $0.1_{\pm0.1}$ | $-0.1_{\pm0.0}$ | $1.4_{\pm1.1}$ | $34.5_{\pm6.2}$ | $-0.1_{\pm0.0}$ | $-0.2_{\pm0.0}$ | $\mathbf{92.1}_{\pm5.0}$ | 100.9 |
| | HUMAN+EXPERT | $0.0_{\pm0.0}$ | $-0.1_{\pm0.0}$ | $-0.2_{\pm0.0}$ | $-0.2_{\pm0.1}$ | $72.3_{\pm5.6}$ | $99.1_{\pm3.1}$ | $0.0_{\pm0.0}$ | $-0.1_{\pm0.0}$ | $\mathbf{102.6}_{\pm2.4}$ | 100.9 |
| KITCHEN | PARTIAL+COMPLETE | $45.5_{\pm0.8}$ | $2.5_{\pm2.2}$ | $5.5_{\pm3.7}$ | $27.3_{\pm2.4}$ | $48.8_{\pm4.0}$ | $45.8_{\pm6.6}$ | $2.8_{\pm0.5}$ | $19.4_{\pm2.1}$ | $\mathbf{53.1}_{\pm5.9}$ | 75.0 |
| | MIXED+COMPLETE | $42.1_{\pm0.5}$ | $2.2_{\pm1.7}$ | $3.1_{\pm2.6}$ | $13.3_{\pm1.4}$ | $\mathbf{50.6}_{\pm1.7}$ | $20.3_{\pm6.3}$ | $1.5_{\pm0.8}$ | $6.7_{\pm2.0}$ | $48.9_{\pm7.3}$ | 75.0 |
| **Average** | | 26.4 | 2.9 | 21.2 | 32.4 | 56.6 | 78.8 | 19.5 | 9.2 | **94.1** | |

*Table 1.* Comparison with other baselines in MuJoCo, Adroit, and FrankaKitchen. The results are normalized score in mean and standard error.

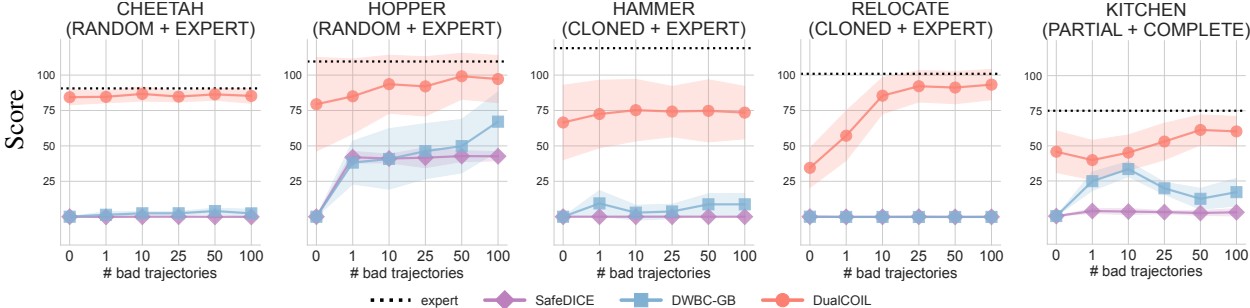

*Figure 1.* Effect of bad dataset size $\mathcal{B}^B$ on performance: Results, averaged over 5 seeds and reported as normalized scores, show that our method effectively leverages increasing numbers of bad trajectories, whereas baselines such as SafeDICE and DWBC-GB fail to do so.

Appendix.

**Evaluation Metrics.** We evaluate all methods using five training seeds. For each seed, we collect the results from the last 10 evaluations (each evaluation consist 10 different environment seeds), then aggregate all evaluations across seeds to compute the mean and standard deviation, which reflect the converged performance of each method. Across all experiments, we report the normalized score commonly used in D4RL tasks $\left(\text{Normalized Score} = \frac{\text{Score} - \text{Random Score}}{\text{Expert Score} - \text{Random Score}}\right)$. This normalization provides a consistent performance measure across different environments.

## 6.2. Main Comparison

To answer Question **(Q1)**, we present a comprehensive comparison between our method and existing baselines across 18 different datasets, as shown in Table 1. First, both BC-MIX and BC-G fail to achieve satisfactory performance across tasks. When learning from the good dataset $\mathcal{B}^G$ and the unlabeled dataset $\mathcal{B}^{\text{MIX}}$, methods like SMODICE and ILID perform reasonably well on the four MuJoCo locomotion tasks (CHEETAH, ANT, HOPPER, WALKER) but completely fail on the five hand manipulation tasks. In contrast, Re-COIL and our method variant (DualCOIL-G) are able to successfully learn in both locomotion and manipulation tasks, demonstrating more robust generalization.

In the setting that incorporates additional low-quality data $\mathcal{B}^B$, SafeDICE shows similar performance to SMODICE and ILID—again failing on the manipulation tasks. Furthermore, DWBC-GB fails to learn entirely, highlighting that a naive adaptation for leveraging poor-quality data can harm the learning process. These results suggest that incorporating bad data $\mathcal{B}^B$ introduces new challenges, and that effectively utilizing such data requires a carefully designed algorithm grounded in strong theoretical principles. Overall, our method successfully leverages the bad dataset $\mathcal{B}^B$ and consistently outperforms all other baselines across both locomotion and manipulation tasks.

## 6.3. Effect of Number of Bad Demonstrations

To answer question **(Q2)**, we investigate the impact of the size of the undesirable (bad) dataset on methods designed to learn from bad data. Specifically, we gradually increase the size of the bad dataset $\mathcal{B}^B$ and evaluate how the performance of each algorithm is affected. The experimental results are presented in Figure 1. Overall, SafeDICE fails to effectively utilize the bad demonstrations, while DWBC-GB is only able to learn in the HOPPER task. In contrast, our method demonstrates strong scalability with respect to the size of the bad dataset, maintaining good performance even when provided with as few as a single bad trajectory.

## 6.4. Comparison with Offline RL Methods

To answer question **(Q3)**, We evaluate whether the implicit reward $\Psi(s, a)$ can be effectively utilized by existing offline RL methods. Specifically, we compare Dual-COIL with several state-of-the-art algorithms, including OPTIDICE (Lee et al., 2021) (as a representative DICE-based method), IQL (Kostrikov et al., 2021), XQL (Garg et al., 2023), and F-DVL (Sikchi et al., 2024), all trained using $\Psi(s, a)$ as the reward. Results are shown in Figure 2.

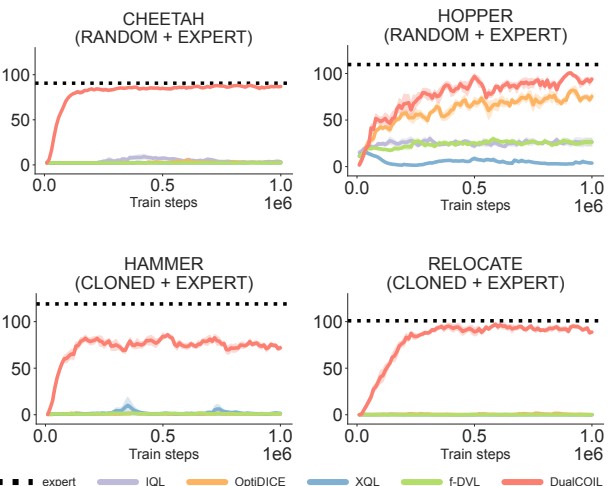

*Figure 2.* Comparison with offline RL methods using the reward $\Psi(s, a)$.

Unlike Q-learning–based methods (e.g., IQL and XQL), which optimize only $\Psi(s, a)$, DualCOIL directly optimizes the full KL-regularized objective in Proposition 4.2. Compared to KL-regularized baselines such as ReCOIL and OptiDICE, DualCOIL adopts a stronger DualRL-inspired framework and introduces (i) a stable approximation of the objective (Proposition 4.4) and (ii) a Q-weighted behavioral cloning update (Proposition 5.1). Ablation studies in Appendices D.4 and D.12 confirm that these components enable DualCOIL to better exploit the implicit reward and outperform direct offline RL baselines.

## 6.5. Sensitivity Analysis of $\alpha$

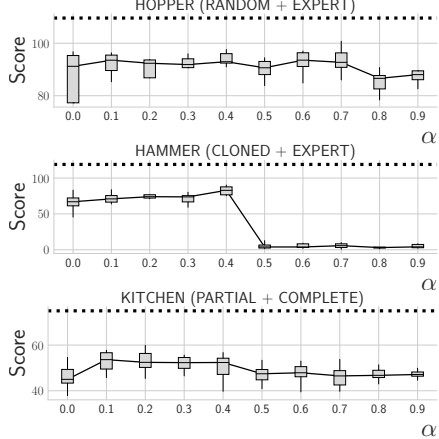

*Figure 3.* Sensitivity analysis on the trade-off parameter $\alpha$.

In our framework, we introduce a hyperparameter $0 \leq \alpha < 1$ to control the contribution of the bad-data objective, addressing question **(Q4)**. To evaluate the sensitivity of our method to $\alpha$, we vary its value and observe the resulting performance, as shown in Figure 3.

Although $\alpha$ has a noticeable effect on performance, our method remains robust over a broad range of values, achieving optimal performance within this interval. The task-specific $\alpha$ values used in our experiments are reported in Appendix C.4.

## 7. Conclusion

We introduced a new offline imitation learning framework that leverages both expert and explicitly undesirable demonstrations. By formulating the learning objective as the difference of KL divergences over visitation distributions, we capture informative contrasts between good and bad behaviors. While the resulting DC program is generally non-convex, we establish conditions under which it becomes convex, leading to a practical, stable, and non-adversarial training procedure. Our unified approach to handling both expert and undesirable demonstrations yields superior performance across a range of offline imitation learning benchmarks, setting a new standard for learning from contrasting behaviors.

**Limitations and Future Work.** Our theoretical analysis focuses on the regime $\alpha \leq 1$, where the learning objective admits a convex structure and can be optimized with strong guarantees. Extending the framework beyond this regime would involve handling non-convex objectives and is an interesting direction for future work. In addition, we assume that the undesirable dataset consists of suboptimal rather than strictly catastrophic behaviors. Extending our approach to safety-critical settings would require incorporating ideas from hard-constrained or safe reinforcement learning, which we leave for future investigation.

## Impact Statement

This paper advances the field of machine learning by proposing an offline imitation learning framework that leverages both positive and negative demonstrations. The method may benefit applications such as robotics, autonomous systems, and decision-support tools where learning from imperfect historical data is common.

As with other imitation learning and reinforcement learning approaches, potential risks include misuse due to biased or low-quality data or deployment in safety-critical settings without sufficient validation. These concerns are not unique to our method and are shared by many data-driven learning systems. We encourage careful consideration of data quality and deployment contexts in practical applications.

Overall, this work makes a general methodological contribution and does not introduce new ethical concerns beyond those associated with existing offline learning methods.

## Acknowledgements

This research is supported by the Singapore International Graduate Award (SINGA) Scholarship from the Agency for Science, Technology and Research, Singapore and the research award awarded to Pradeep Varakantham by Google.

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

# Appendix

This appendix includes the following materials:

**Missing Proofs:** Proofs omitted from the main paper are provided in Appendix A.

**Additional Discussions:** We also provide some additional discussions about our method:

- Derivation of the Q-learning Objective (Appendix B.1)

- A Note on DualCOIL under other $f$-Divergences (Appendix B.2)

- Possible Failure Modes in Avoiding Bad Demonstrations (Appendix B.3)

- Tightness of the Surrogate Lower Bound and the Role of Regularization (Appendix B.4)

**Experimental Details:** We describe the following aspects in detail:

- Full pseudocode (Appendix C.1)

- Dataset construction (Appendix C.2)

- Baseline implementations (Appendix C.3)

- Hyperparameter selections (Appendix C.4)

- Computational resources (Appendix C.5)

**Additional Experiments:** We further present supplementary results:

- Comparison between the original exponential-form objective function $L(Q, \pi)$ and the lower-bound surrogate $\widetilde{L}(Q, \pi)$ (Appendix D.12)

- Effect of the size of the bad dataset (Appendix D.1)

- Effect of the number of expert demonstrations in the good dataset $\mathcal{B}^G$ (Appendix D.2)

- Discussion: How many bad trajectories in $\mathcal{B}^B$ are sufficient to replace a good trajectory in $\mathcal{B}^G$ for DualCOIL? (Appendix D.3)

- Comparison of advantage-weighted BC and Q-weighted BC for policy extraction (Appendix D.4)

- Performance across varying quality levels of the unlabeled dataset $\mathcal{B}^{\text{Mix}}$ (Appendix D.5)

- Comparison with adapted offline reinforcement learning methods (Appendix D.8)

- Discussion: distribution-matching vs. preference-based approaches (Appendix D.9)

- Additional comparison with "avoid-bad-only" baselines (Appendix D.10)

- Ablations and experiments with $\alpha > 1$ (Appendix D.11)

- Sensitivity analysis of $\beta$ (Appendix D.13)

- The performance different betweenExtreme-V and Log-sum-exp for Value Function Estimation (Appendix D.14)

- How DualCOIL perform in the Safety tasks (Appendix D.15)

**Stress Tests:** We conduct additional stress-test experiments by:

- Increasing the proportion of bad data in the unlabeled dataset $\mathcal{B}^{\text{Mix}}$ to very high levels (Appendix D.6)

- Generating more bad data to enlarge the bad dataset $\mathcal{B}^B$ (Appendix D.7)

# A. Missing Proofs

This section provides proofs of the theoretical results that are omitted from the main paper.

**Proposition 4.1:** *If $\alpha \leq 1$, then the objective function $f(d^{\pi}) = D_{KL}(d^{\pi} \parallel d^G) - \alpha\, D_{KL}(d^{\pi} \parallel d^B)$ is convex in $d^{\pi}$.*

*Proof.* We write the objective function as:

$$
\begin{aligned}
f(d^{\pi}) &= \sum_{(s,a)\sim d^{\pi}} \log \frac{d^{\pi}(s,a)}{d^G(s,a)} - \alpha \sum_{(s,a)\sim d^{\pi}} \log \frac{d^{\pi}(s,a)}{d^B(s,a)} \\
&= \sum_{s,a} (1-\alpha) d^{\pi}(s,a) \log d^{\pi}(s,a) + d^{\pi}(s,a)(\alpha d^B(s,a) - d^G(s,a)) \qquad (10)
\end{aligned}
$$

We can see that the first term is convex in $d^{\pi}$ since $\alpha \leq 1$ and $d^{\pi}(s,a) \log d^{\pi}(s,a)$ is convex in $d^{\pi}$. Moreover, the second term is linear in $d^{\pi}$. This implies that $f(d^{\pi})$ is convex in $\pi$ if $\alpha \leq 1$, as desired.

$\square$

**Proposition 4.2:** *The objective function in* (2) *can be written as:* $f(d, \pi) = (1-\alpha)D_{KL}(d||d^U) - \mathbb{E}_{(s,a)\sim d}[\Psi(s,a)]$, *where* $\Psi(s,a) = \log \frac{d^G(s,a)}{d^U(s,a)} - \alpha \log \frac{d^B(s,a)}{d^U(s,a)}$.

*Proof.* We can expand the objective function as:

$$
f(d, \pi) = \mathbb{E}_{(s,a)\sim d}\left[\log \frac{d(s,a)}{d^G(s,a)}\right] - \alpha\, \mathbb{E}_{(s,a)\sim d}\left[\log \frac{d(s,a)}{d^B(s,a)}\right].
$$

We can rewrite the objective using $d^U$ as an intermediate distribution:

$$
\begin{aligned}
f(d, \pi) &= \mathbb{E}_{(s,a)\sim d}\left[\log \frac{d(s,a)}{d^G(s,a)}\right] - \alpha\, \mathbb{E}_{(s,a)\sim d}\left[\log \frac{d(s,a)}{d^B(s,a)}\right] \\
&= \mathbb{E}_{(s,a)\sim d}\left[\log \frac{d(s,a)}{d^U(s,a)} + \log \frac{d^U(s,a)}{d^G(s,a)}\right] - \alpha\, \mathbb{E}_{(s,a)\sim d}\left[\log \frac{d(s,a)}{d^U(s,a)} + \log \frac{d^U(s,a)}{d^B(s,a)}\right] \\
&= (1-\alpha)\, \mathbb{E}_{(s,a)\sim d}\left[\log \frac{d(s,a)}{d^U(s,a)}\right] - \mathbb{E}_{(s,a)\sim d}\left[\Psi(s,a)\right], \\
&= (1-\alpha)D_{KL}(d||d^U) - \mathbb{E}_{(s,a)\sim d}\left[\Psi(s,a)\right]
\end{aligned}
$$

where $\Psi(s,a) = \log \frac{d^G(s,a)}{d^U(s,a)} - \alpha \log \frac{d^B(s,a)}{d^U(s,a)}$. $\square$

**Proposition 4.3:** *Let the surrogate objective be defined as:*

$$
\widetilde{L}(Q, \pi) = (1-\gamma)\, \mathbb{E}_{s\sim p_0}\left[V_Q^{\pi}(s)\right] - \mathbb{E}_{d^U}\left[\delta(s,a)\mathcal{T}^{\pi}[Q](s,a)\right] + (1-\alpha)\mathbb{E}_{d^U}\left[\delta(s,a)\right]. \qquad (11)
$$

*where* $\delta(s,a) = \exp\left(\frac{\Psi(s,a)}{1-\alpha}\right)$. *Then* $\widetilde{L}(Q, \pi)$ *is a lower bound of* $L(Q, \pi)$, *with equality when* $\mathcal{T}^{\pi}[Q](s,a) = 0$ *for all* $(s, a)$.

*Proof.* We first write $L(Q, \pi)$ as:

$$
\begin{aligned}
L(Q, \pi) &= (1 - \gamma)\, \mathbb{E}_{s \sim p_0}\left[ V_Q^{\pi}(s) \right] \\
&\quad + (1 - \alpha)\mathbb{E}_{(s,a) \sim d^U}\left[ \exp\left( \frac{\Psi(s,a) - \mathcal{T}^{\pi}[Q](s,a)}{1 - \alpha} \right) \right] \\
&= (1 - \gamma)\, \mathbb{E}_{s \sim p_0}\left[ V_Q^{\pi}(s) \right] \\
&\quad + (1 - \alpha)\mathbb{E}_{(s,a) \sim d^U}\left[ \exp\left( \frac{\Psi(s,a)}{1 - \alpha} \right) \exp\left( \frac{-\mathcal{T}^{\pi}[Q](s,a)}{1 - \alpha} \right) \right] \\
&= (1 - \gamma)\, \mathbb{E}_{s \sim p_0}\left[ V_Q^{\pi}(s) \right] \\
&\quad + (1 - \alpha)\mathbb{E}_{(s,a) \sim d^U}\left[ \delta(s,a) \exp\left( \frac{-\mathcal{T}^{\pi}[Q](s,a)}{1 - \alpha} \right) \right],
\end{aligned}
$$

where we define $\delta(s,a) := \exp\left( \frac{\Psi(s,a)}{1 - \alpha} \right)$.

Now, we use the inequality $e^t \geq t + 1$ (which follows from the convexity of $e^t$ and is tight at $t = 0$), to obtain:

$$
\exp\left( \frac{-\mathcal{T}^{\pi}[Q](s,a)}{1 - \alpha} \right) \geq -\frac{\mathcal{T}^{\pi}[Q](s,a)}{1 - \alpha} + 1.
$$

Substituting this into the expression for $L(Q, \pi)$, we get:

$$
L(Q, \pi) \geq (1 - \gamma)\, \mathbb{E}_{s \sim p_0}\left[ V_Q^{\pi}(s) \right] + (1 - \alpha)\mathbb{E}_{(s,a) \sim d^U}\left[ \delta(s,a)\left( -\frac{\mathcal{T}^{\pi}[Q](s,a)}{1 - \alpha} + 1 \right) \right] =: \widetilde{L}(Q, \pi).
$$

Equality holds in the inequality $e^t \geq t + 1$ when $t = 0$, which corresponds to $\mathcal{T}^{\pi}[Q](s,a) = 0$. That is, the equality $L(Q, \pi) = \widetilde{L}(Q, \pi)$ holds when the rewards represented by the $Q$-function are zero everywhere. This completes the proof. $\qquad\square$

**Proposition 4.4:** $\widetilde{L}(Q, \pi)$ *is linear in $Q$ and concave in $\pi$. As a result, the max–min optimization $\max_{\pi} \min_Q \widetilde{L}(Q, \pi)$ equivalently reduces to the following non-adversarial problem:*

$$
\min_Q \left\{ \widetilde{L}(Q) = (1 - \gamma)\, \mathbb{E}_{s \sim p_0}\left[ V_Q(s) \right] - \mathbb{E}_{(s,a) \sim d^U}\left[ \exp\left( \frac{\Psi(s,a)}{1 - \alpha} \right) \mathcal{T}[Q](s,a) \right] \right\},
$$

*where the soft value function $V_Q(s)$ is defined as: $V_Q(s) = \beta \log\left( \sum_a \mu^U(a|s) \exp(Q(s,a)/\beta) \right)$, and the soft Bellman residual operator is given by: $\mathcal{T}[Q](s,a) = Q(s,a) - \gamma V_Q(s)$. Moreover $\widetilde{L}(Q)$ is convex in $Q$.*

*Proof.* We first write $\widetilde{L}(Q, \pi)$ as:

$$
\begin{aligned}
\widetilde{L}(Q, \pi) &= (1 - \gamma)\, \mathbb{E}_{s \sim p_0}\left[ V_Q^{\pi}(s) \right] - \mathbb{E}_{(s,a) \sim d^U}\left[ \delta(s,a)\left( Q(s,a) - \gamma \mathbb{E}_{s'}\left[ V_Q^{\pi}(s') \right] \right) \right] \\
&\quad + (1 - \alpha)\mathbb{E}_{(s,a) \sim d^U}\left[ \delta(s,a) \right],
\end{aligned}
$$

where we recall that

$$
V_Q^{\pi}(s) = \mathbb{E}_{a \sim \pi(\cdot|s)}\left[ Q(s,a) - \beta \log \frac{\pi(a \mid s)}{\mu^U(a \mid s)} \right].
$$

Thus, we can observe that $\widetilde{L}(Q, \pi)$ is linear in $Q$.

Moreover, the function $V_Q^{\pi}(s)$ is concave in $\pi$, since it is composed of the expectation over a linear function of $\pi$ (through $Q(s,a)$) and the negative entropy-regularized KL-divergence term, which is convex in $\pi$ and thus its negative is concave. That is,

$$
V_Q^{\pi}(s) = \mathbb{E}_{a \sim \pi(\cdot|s)}\left[ Q(s,a) - \beta \log \frac{\pi(a \mid s)}{\mu^U(a \mid s)} \right]
$$

is concave in $\pi$.

Furthermore, since $\delta(s, a) > 0$, the coefficients associated with $V_Q^\pi(s)$ in $\widetilde{L}(Q, \pi)$ are non-negative. This implies that the entire function $\widetilde{L}(Q, \pi)$ is concave in $\pi$.

Now, since $\widetilde{L}(Q, \pi)$ is concave in $\pi$ and linear in $Q$, we can apply the minimax theorem to swap the order of the max and min:

$$\max_\pi \min_Q \widetilde{L}(Q, \pi) = \min_Q \max_\pi \widetilde{L}(Q, \pi).$$

This holds because the function $\widetilde{L}(Q, \pi)$ satisfies the standard conditions of the minimax theorem: it is concave in $\pi$, convex (in fact, linear) in $Q$, and the optimization domains are convex.

Next, observe that in $\widetilde{L}(Q, \pi)$, the variable $\pi$ only appears through the term $V_Q^\pi(s)$, and all coefficients multiplying $V_Q^\pi(s)$ are non-negative. Therefore, maximizing $\widetilde{L}(Q, \pi)$ over $\pi$ is equivalent to maximizing $V_Q^\pi(s)$ for each state $s$ independently. That is,

$$\max_\pi \widetilde{L}(Q, \pi) \equiv \max_\pi \sum_s c(s) V_Q^\pi(s),$$

for some non-negative coefficients $c(s) \geq 0$, which implies it suffices to solve $\max_\pi V_Q^\pi(s)$ pointwise.

Recall the definition:

$$V_Q^\pi(s) = \mathbb{E}_{a \sim \pi(\cdot|s)} \left[ Q(s, a) - \beta \log \frac{\pi(a \mid s)}{\mu^U(a \mid s)} \right].$$

The inner maximization over $\pi(\cdot \mid s)$ is a standard entropy-regularized problem, and the optimal policy has the closed-form solution:

$$\pi^*(a \mid s) = \frac{\mu^U(a \mid s) \exp\left( \frac{Q(s,a)}{\beta} \right)}{\sum_{a'} \mu^U(a' \mid s) \exp\left( \frac{Q(s,a')}{\beta} \right)}.$$

This is a weighted softmax over $Q(s, a)$ values, using the baseline distribution $\mu^U(a \mid s)$ as the reference. Substituting this back into $V_Q^\pi(s)$ yields the closed-form maximized value:

$$\max_\pi V_Q^\pi(s) = \beta \log \left( \sum_a \mu^U(a \mid s) \exp\left( \frac{Q(s, a)}{\beta} \right) \right).$$

Thus:

$$\min_Q \max_\pi \widetilde{L}(Q, \pi) = \min_Q \widetilde{L}(Q)$$

where

$$\widetilde{L}(Q) = (1 - \gamma) \mathbb{E}_{s \sim p_0} [V_Q(s)] - \mathbb{E}_{(s,a) \sim d^U} \left[ \exp\left( \frac{\Psi(s, a)}{1 - \alpha} \right) (Q(s, a) - \gamma \mathbb{E}_{s'} [V_Q(s')]) \right],$$

and

$$V_Q(s) = \beta \log \sum_a \mu^U(a \mid s) \exp\left( \frac{Q(s, a)}{\beta} \right).$$

We can now see that $\widetilde{L}(Q)$ is convex in $Q$, due to the following reasons:

- The function $Q(s, a) \mapsto \log \sum_a \mu^U(a \mid s) \exp\left( \frac{Q(s,a)}{\beta} \right)$ is a softmax (log-sum-exp), which is convex.

- $V_Q(s)$, being a composition of a convex function with an affine transformation, is convex in $Q$.

- Expectations over convex functions (e.g., $\mathbb{E}_{s \sim p_0}[V_Q(s)]$, $\mathbb{E}_{s'}[V_Q(s')]$) preserve convexity.

- The remaining terms in $\widetilde{L}(Q)$, such as $Q(s, a)$, appear linearly and thus preserve convexity.

Hence, the overall objective $\widetilde{L}(Q)$ is convex in $Q$, which completes the proof.

$\square$

**Proposition 5.1** *The following Q-weighted behavior cloning (BC) objective yields the same optimal policy as the original advantage-weighted BC formulation in* (8)*:*

$$\max_{\pi} \sum_{(s,a)\sim\mathcal{B}^U} \exp\left(\frac{1}{\beta}Q(s,a)\right) \log \pi(a \mid s). \tag{12}$$

*Proof.* The Q-weighted BC objective can be written as:

$$\max_{\pi} \sum_{(s,a)} \mu^U(s,a) \exp\left(\frac{1}{\beta}Q(s,a)\right) \log \pi(a \mid s).$$

This represents a weighted maximum likelihood objective, where the weights are shaped by the exponential of the Q-values. For each state $s$, the optimal solution $\pi^*(a \mid s)$ is given by:

$$\pi^*(a \mid s) = \frac{\mu^U(s,a) \exp\left(\frac{1}{\beta}Q(s,a)\right)}{\sum_{a'} \mu^U(s,a') \exp\left(\frac{1}{\beta}Q(s,a')\right)}.$$

Moreover, we recall that:

$$V^Q(s) = \beta \log \left(\sum_{a'} \mu^U(s,a') \exp\left(\frac{1}{\beta}Q(s,a')\right)\right),$$

which allows us to express the optimal policy in terms of the advantage $Q(s,a) - V^Q(s)$ as:

$$\pi^*(a \mid s) = \mu^U(s,a) \exp\left(\frac{1}{\beta}(Q(s,a) - V^Q(s))\right).$$

This is precisely the optimal policy corresponding to the advantage-weighted BC objective defined in (8). This completes the proof.

$\square$

# B. Additional Discussions

## B.1. Derivation of the Q-learning Objective

We provide a detailed derivation of the Q-learning objective presented in Section 4.1 of the main paper, starting from the primal objective in Proposition 4.2. The optimization problem can be written as:

$$\min_{d,\pi} (1-\alpha)D_{\mathrm{KL}}(d \,||\, d_U) - \mathbb{E}_{(s,a)\sim d}[\Psi(s,a)]$$
$$\text{s.t.} \quad d(s,a) = (1-\gamma)p_0(s)\pi(a|s) + \gamma\pi(a|s)\sum_{s',a'} T(s|s',a')d(s',a'), \tag{13}$$

where $\Psi(s,a) = \log \frac{d_G(s,a)}{d_U(s,a)} - \alpha \log \frac{d_B(s,a)}{d_U(s,a)}$. This objective is of the same form as the primal regularized RL problem analyzed in Sikchi et al. (2024), with reward $r(s,a) = \Psi(s,a)$, reference distribution $d_{\mathrm{ref}} = d_U$, and regularization weight $(1-\alpha)$. Following the derivation in DualRL, we introduce a Lagrange multiplier $Q(s,a)$ for the occupancy-flow constraint and form the Lagrangian:

$$\mathcal{L}(d,\pi,Q) = (1-\alpha)\mathbb{E}_d\left[\log \frac{d(s,a)}{d_U(s,a)}\right] - \mathbb{E}_d[\Psi(s,a)]$$
$$+ \mathbb{E}_d\left[Q(s,a) - \gamma\mathbb{E}_{T,\pi}[Q(s',a')]\right] - (1-\gamma)\mathbb{E}_{p_0,\pi}[Q(s,a)]. \tag{14}$$

Minimizing (14) with respect to $d(s,a)$ pointwise gives

$$\frac{\partial \mathcal{L}}{\partial d(s,a)} = 0 \quad \Rightarrow \quad \log \frac{d^*(s,a)}{d_U(s,a)} = \frac{1}{1-\alpha}\left[\Psi(s,a) + \gamma\mathbb{E}_{T,\pi}[Q(s',a')] - Q(s,a)\right] + c,$$

which leads to the optimal occupancy distribution

$$d^*(s,a) \propto d_U(s,a) \exp\left( \frac{\Psi(s,a) + \gamma \, \mathbb{E}_{T,\pi}[Q(s',a')] - Q(s,a)}{1-\alpha} \right). \tag{15}$$

Substituting (15) back into the Lagrangian eliminates the dependence on $d$, yielding the following dual optimization problem:

$$\max_{\pi} \min_{Q} \left[ (1-\gamma) \, \mathbb{E}_{p_0,\pi}[Q(s,a)] \right.$$
$$\left. + (1-\alpha) \, \mathbb{E}_{(s,a) \sim d_U} \left[ \exp\left( \frac{\Psi(s,a) + \gamma \, \mathbb{E}_{T,\pi}[Q(s',a')] - Q(s,a)}{1-\alpha} \right) \right] \right]. \tag{16}$$

Equation 16 corresponds exactly to the Q-learning formulation presented in Eq. 3.

Here we note that we adopt the DualRL framework because it provides a principled and unified treatment of regularized reinforcement learning objectives. In particular, DualRL optimizes a KL-regularized objective that naturally arises from our derivation in Proposition 4.2, ensuring theoretical consistency between the recovered reward model and the downstream policy optimization. Moreover, DualRL has demonstrated strong empirical performance across diverse benchmarks and was highlighted as an ICLR 2024 spotlight presentation. Its stability and efficiency make it a more advanced and reliable choice compared to earlier DICE-based estimators or Q-learning variants such as OptDICE or XQL. Integrating DualRL therefore enables us to fully leverage our implicit reward formulation while benefiting from a state-of-the-art optimization framework.

## B.2. A Note on DualCOIL under other $f$-Divergences

We note that the convexity stated in Proposition 4.1 does not hold under arbitrary $f$-divergences, even under the same assumptions. To illustrate this, consider the following objective defined using an $f$-divergence:

$$F(d^\pi) = D_f(d^\pi \parallel d^G) - \alpha \, D_f(d^\pi \parallel d^B),$$

which can be written as:

$$F(d^\pi) = \sum_{(s,a)} d^G(s,a) f\left( \frac{d^\pi(s,a)}{d^G(s,a)} \right) - \alpha \, d^B(s,a) f\left( \frac{d^\pi(s,a)}{d^B(s,a)} \right).$$

Observe that each term

$$d^G(s,a) f\left( \frac{d^\pi(s,a)}{d^G(s,a)} \right) - \alpha \, d^B(s,a) f\left( \frac{d^\pi(s,a)}{d^B(s,a)} \right)$$

is not necessarily convex for any $\alpha > 0$. Whether this expression is convex depends on the values of $d^G(s,a)$ and $d^B(s,a)$. In particular, if $d^G(s,a) = 0$—i.e., the state-action pair $(s,a)$ is never visited by the expert policy—then the term may become concave. Therefore, in general, the objective $F(d^\pi)$ defined under an $f$-divergence is not convex in $d^\pi$ for arbitrary choices of $\alpha$. Thus, the standard Lagrangian duality cannot be applied. For this reason, the KL divergence appears to be an ideal choice for our problem of learning from both expert and undesirable demonstrations.

## B.3. Possible Failure Modes in Avoiding Bad Demonstrations

A known challenge in pushing the policy distribution away from the mean of the bad dataset is the potential emergence of new undesirable behaviors not covered by the dataset—often referred to as a "whack-a-mole" problem. If the bad dataset is incomplete, the policy may still converge to harmful behaviors. In this context, the primal objective $D_{KL}(d^\pi \parallel d^G) - D_{KL}(d^\pi \parallel d^B)$ highlights the importance of expert demonstrations, as safe learning requires that the influence of expert behavior outweighs that of bad behavior.

Our framework addresses this by prioritizing imitation of expert behavior whenever available, while using bad demonstrations only to avoid clearly undesirable actions. Thus, expert data anchors the policy, and bad data serves as a supplementary signal rather than requiring exhaustive coverage of all failure modes. This mitigates the "whack-a-mole" issue by ensuring the policy remains primarily guided by expert behavior.

Naturally, when both expert and bad demonstrations are scarce, policy learning becomes difficult—a limitation shared by most IL approaches. Nonetheless, our experiments show that the method is robust and consistently outperforms baselines, even with only limited expert data.

### B.4. Tightness of the Surrogate Lower Bound and the Role of Regularization

In our algorithm, we adopt the surrogate objective $\widetilde{L}(Q, \pi)$ as a tractable lower bound of the true training objective $L(Q, \pi)$. A natural concern is: *How tight is the lower-bound objective $\widetilde{L}(Q, \pi)$ compared to the original objective $L(Q, \pi)$?* While the gap between $L(Q, \pi)$ and its surrogate can be nontrivial—reflecting the difference between the exponential function and its linear approximation—this does not undermine its effectiveness. The surrogate offers tractability while still guiding the optimization of $Q$ and $\pi$ in a direction consistent with maximizing the original objective, since both $e^x$ and $x + 1$ share the same monotonicity. Appendix D.12 provides a detailed comparison, and ablation studies confirm that the surrogate leads to significantly improved training performance.

Another question is whether DualCOIL benefits primarily from the implicit regularization within $L(Q, \pi)$ rather than being a faithful proxy for the original $D_{KL}(d^\pi \,\|\, d^G) - D_{KL}(d^\pi \,\|\, d^B)$ objective. In practice, this regularization mainly stabilizes training by preventing extreme $Q$-values, a technique also found in baselines such as SafeDICE and DWBC. However, regularization alone does not enable meaningful learning from both expert and undesirable datasets. The superior performance of DualCOIL arises instead from the structure of $L(Q, \pi)$ itself, which is grounded in the original KL-divergence formulation.

# C. Experiment Settings

## C.1. Full Pseudo Code

The detailed implementation are provided in Algorithm 2.

---

**Algorithm 2** DualCOIL: Offline Imitation Learning from Contrasting Behaviors (full)

---

**Require:** Good dataset $\mathcal{B}_G$, Bad dataset $\mathcal{B}_B$, unlabeled dataset $\mathcal{B}_U$
**Require:** Hyperparameters: $\alpha \in [0, 1)$, $\beta$, $\gamma$, $N_\mu$, $N$, target update rate $\tau$, batch size $B$
 1: Initialize networks: $Q_{w_q}(s, a)$, $V_{w_v}(s)$, $\pi_\theta(a|s)$, classifiers $c_{w_G}^G(s, a)$, $c_{w_B}^B(s, a)$
 2: Initialize target Q-network: $Q_{\text{target}} \leftarrow Q_{w_q}$
 3:
 4: Step 1: Estimate occupancy ratios
 5: **for** $i = 1$ to $N_\mu$ **do**
 6:    Sample batch $\{(s_i^G)'\}_{i=1}^B \sim \mathcal{B}_G$; $\{(s_i^B)'\}_{i=1}^B \sim \mathcal{B}_B$; $\{(s_i^U)'\}_{i=1}^B \sim \mathcal{B}_U$
 7:    Update $c_{w_G}^G$ by maximizing the objective in (6).
 8:    Update $c_{w_B}^B$ by maximizing an analogous objective to (6) for the bad dataset.
 9: **end for**
10:
11: Step 2: Calculate $\Psi$ function
12: Calculate $\Psi(s, a) = \log\left(\frac{c_{w_G}^G(s')}{1 - c_{w_G}^G(s')}\right) - \alpha \log\left(\frac{c_{w_B}^B(s')}{1 - c_{w_B}^B(s')}\right)$.
13:
14: Step 3: Train Q, V, and Policy
15: **for** $i = 1$ to $N$ **do**
16:    Sample batch $\{(s_i, a_i, s_i', \Psi_i)\}_{i=1}^B \sim \mathcal{B}_U$
17:    **Q-Update:** Minimize the objective $\tilde{L}(Q_{w_q}|V_{w_v}) + \frac{1}{2}(Q_{w_q}(s_i, a_i) - \gamma V_{w_v}(s_i'))^2$ .
18:                                                           (reference: $\tilde{L}(Q|V)$ from Sec 5/ Eq (7))
19:    **V-Update:** Minimize the Extreme-V objective:

$$\min_{w_v} \frac{1}{B} \sum_{i=1}^B \left[ \exp\left(\frac{Q_{\text{target}}(s_i, a_i) - V_{w_v}(s_i)}{\beta}\right) - \frac{Q_{\text{target}}(s_i, a_i) - V_{w_v}(s_i)}{\beta} - 1 \right].$$

20:    **Policy Update:** Maximize the policy by using Q-weighted Behavior Cloning.
21:                                                                    (reference: Sec 5/ Eq (9))
22:    **Target Q-Update:** Soft update: $Q_{\text{target}} \leftarrow \tau Q_{w_q} + (1 - \tau) Q_{\text{target}}$
23: **end for**

---

## C.2. Dataset Construction

From the official D4RL dataset we use three different domains:

- MuJoCo Locomotion[CHEETAH,ANT,HOPPER,WALKER] with three types of dataset:
  - EXPERT
  - MEDIUM
  - RANDOM

- Adroit [PEN,HAMMER,DOOR,RELOCATE] with three types of dataset:
  - EXPERT
  - HUMAN
  - CLONED

- FrankaKitchen [KITCHEN] with three types of dataset:
  - COMPLETE
  - MIXED
  - PARTIAL

Following the approach of (Sikchi et al., 2024), we also provide several combinations across all three domains, as shown in Table 2. Notably, the unlabeled dataset $\mathcal{B}^{\text{MIX}}$ is constructed by combining the entire suboptimal dataset with the expert dataset, resulting in an overlap between $\mathcal{B}^B$ and $\mathcal{B}^{\text{MIX}}$. Nevertheless, this setup is practical: given an good dataset $\mathcal{B}^G$ and an unlabeled dataset $\mathcal{B}^{\text{MIX}}$, users can randomly sample trajectories and assign them to either $\mathcal{B}^G$ or $\mathcal{B}^B$ without the need for any additional external data.

| Task | Unlabeled name | $\mathcal{B}^G$ | $\mathcal{B}^B$ | $\mathcal{B}^{\text{MIX}}$ |
|---|---|---|---|---|
| CHEETAH | RANDOM+EXPERT | 1 EXPERT | 10 RANDOM | Full RANDOM+30 EXPERT |
| | MEDIUM+EXPERT | 1 EXPERT | 10 MEDIUM | Full MEDIUM+30 EXPERT |
| ANT | RANDOM+EXPERT | 1 EXPERT | 10 RANDOM | Full RANDOM+30 EXPERT |
| | MEDIUM+EXPERT | 1 EXPERT | 10 MEDIUM | Full MEDIUM+30 EXPERT |
| HOPPER | RANDOM+EXPERT | 1 EXPERT | 10 RANDOM | Full RANDOM+30 EXPERT |
| | MEDIUM+EXPERT | 1 EXPERT | 10 MEDIUM | Full MEDIUM+30 EXPERT |
| WALKER | RANDOM+EXPERT | 1 EXPERT | 10 RANDOM | Full RANDOM+30 EXPERT |
| | MEDIUM+EXPERT | 1 EXPERT | 10 MEDIUM | Full MEDIUM+30 EXPERT |
| PEN | CLONED+EXPERT | 1 EXPERT | 25 CLONED | Full CLONED+100 EXPERT |
| | HUMAN+EXPERT | 1 EXPERT | 25 HUMAN | Full HUMAN+100 EXPERT |
| HAMMER | CLONED+EXPERT | 1 EXPERT | 25 CLONED | Full CLONED+100 EXPERT |
| | HUMAN+EXPERT | 1 EXPERT | 25 HUMAN | Full HUMAN+100 EXPERT |
| DOOR | CLONED+EXPERT | 1 EXPERT | 25 CLONED | Full CLONED+100 EXPERT |
| | HUMAN+EXPERT | 1 EXPERT | 25 HUMAN | Full HUMAN+100 EXPERT |
| RELOCATE | CLONED+EXPERT | 1 EXPERT | 25 CLONED | Full CLONED+100 EXPERT |
| | HUMAN+EXPERT | 1 EXPERT | 25 HUMAN | Full HUMAN+100 EXPERT |
| KITCHEN | PARTIAL+COMPLETE | 1 COMPLETE | 25 PARTIAL | Full PARTIAL+1 COMPLETE |
| | MIXED+COMPLETE | 1 COMPLETE | 25 MIXED | Full MIXED+1 COMPLETE |

*Table 2.* **Dataset Construction.** The numbers in Table 2 indicate the number of trajectories drawn from each corresponding dataset. For the KITCHEN task, we follow the setting of (Sikchi et al., 2024), where only a single trajectory from the COMPLETE dataset is included in $\mathcal{B}^{\text{MIX}}$.

## C.3. Baselines Implementation

We compare our method against several established baselines. For methods with publicly available code, we utilized their official implementations without algorithmic modifications.

### C.3.1. BEHAVIOR CLONING (BC)

We employ the standard Behavior Cloning (BC) objective, which aims to minimize the negative log-likelihood of the demonstrated actions under the learned policy:

$$\min_{\pi} -\mathbb{E}_{(s,a)\sim\mathcal{B}} \log \pi(a \mid s), \tag{17}$$

where $\mathcal{B}$ denotes the dataset of state-action pairs. Specifically, $\mathcal{B}$ corresponds to $\mathcal{B}^{\text{MIX}}$ in the case of BC-MIX, or $\mathcal{B}^G$ for BC-G.

### C.3.2. OTHER BASELINES WITH OFFICIAL IMPLEMENTATIONS

For the following baselines, we used their official, unmodified implementations:

- **SMODICE** (Ma et al., 2022): Applied to both the good dataset ($\mathcal{B}^G$) and the mixed dataset ($\mathcal{B}^{\text{MIX}}$). The official code is available at [GitHub].

- **ILID** (Yue et al., 2024): Applied to $\mathcal{B}^G$ and $\mathcal{B}^{\text{MIX}}$. The official code is available at [GitHub].

- **ReCOIL** (Sikchi et al., 2024): Applied to $\mathcal{B}^G$ and $\mathcal{B}^{\text{MIX}}$. The official code is available at [GitHub].

- **SafeDICE** (Jang et al., 2024): Applied to the bad dataset ($\mathcal{B}^B$) and the mixed dataset ($\mathcal{B}^{\text{MIX}}$). The official code is available at [GitHub].

### C.3.3. DWBC-GB

DWBC-GB is our adaptation of DWBC (Xu et al., 2022) (original official implementation: [GitHub]). While the original DWBC is designed for scenarios involving $\mathcal{B}^G$ and $\mathcal{B}^{\text{MIX}}$, our modified version, DWBC-GB, is extended to handle all three dataset types: $\mathcal{B}^G$, $\mathcal{B}^B$, and $\mathcal{B}^{\text{MIX}}$.

This adaptation involves training two discriminators: $c^G$ for good data and $c^B$ for bad data. Their respective loss functions are:

$$\begin{aligned}
L_{c^G} = \eta\, \mathbb{E}_{(s,a)\sim\mathcal{B}^G}&[-\log c^G(s,a,\log\pi(a|s))] \\
&+ \mathbb{E}_{(s,a)\sim\mathcal{B}^{\text{MIX}}}[-\log(1-c^G(s,a,\log\pi(a|s)))] \\
&- \eta\, \mathbb{E}_{(s,a)\sim\mathcal{B}^G}[-\log(1-c^G(s,a,\log\pi(a|s)))],
\end{aligned} \tag{18}$$

$$\begin{aligned}
L_{c^B} = \eta\, \mathbb{E}_{(s,a)\sim\mathcal{B}^B}&[-\log c^B(s,a,\log\pi(a|s))] \\
&+ \mathbb{E}_{(s,a)\sim\mathcal{B}^{\text{MIX}}}[-\log(1-c^B(s,a,\log\pi(a|s)))] \\
&- \eta\, \mathbb{E}_{(s,a)\sim\mathcal{B}^B}[-\log(1-c^B(s,a,\log\pi(a|s)))].
\end{aligned} \tag{19}$$

The policy $\pi$ is then learned by minimizing the objective:

$$\begin{aligned}
\min_{\pi} \Bigg( \mathbb{E}_{(s,a)\sim\mathcal{B}^G} &\left[ -\log\pi(a|s) \cdot \left( \alpha - \frac{\eta}{c(s,a)\,(1-c(s,a))} \right) \right] \\
&+ \mathbb{E}_{(s,a)\sim\mathcal{B}^{\text{MIX}}} \left[ -\log\pi(a|s) \cdot \frac{1}{1-c(s,a)} \right] \Bigg),
\end{aligned} \tag{20}$$

where $c(s,a) = c^G(s,a) - c^B(s,a)$. (Note: $\eta$ and $\alpha$ are hyperparameters.)

## C.4. Hyper Parameters

Our method features two primary hyperparameters: $\alpha$ (weighting for balancing positive and negative samples) and $\beta$ (Extreme-V update). Sections 6.5, D.11, and D.13 present ablation studies detailing the sensitivity to these parameters.

Specific parameters for all tasks are provided in Table 3 below:

| Task | Unlabeled name | $\alpha$ | $\beta$ |
|---|---|---|---|
| CHEETAH | RANDOM+EXPERT | 0.6 | 20.0 |
|  | MEDIUM+EXPERT | 0.6 | 15.0 |
| ANT | RANDOM+EXPERT | 0.6 | 15.0 |
|  | MEDIUM+EXPERT | 0.6 | 15.0 |
| HOPPER | RANDOM+EXPERT | 0.4 | 30.0 |
|  | MEDIUM+EXPERT | 0.4 | 30.0 |
| WALKER | RANDOM+EXPERT | 0.6 | 20.0 |
|  | MEDIUM+EXPERT | 0.6 | 20.0 |
| PEN | CLONED+EXPERT | 0.4 | 15.0 |
|  | HUMAN+EXPERT | 0.4 | 10.0 |
| HAMMER | CLONED+EXPERT | 0.2 | 10.0 |
|  | HUMAN+EXPERT | 0.6 | 20.0 |
| DOOR | CLONED+EXPERT | 0.4 | 15.0 |
|  | HUMAN+EXPERT | 0.4 | 10.0 |
| RELOCATE | CLONED+EXPERT | 0.4 | 30.0 |
|  | HUMAN+EXPERT | 0.8 | 3.0 |
| KITCHEN | PARTIAL+COMPLETE | 0.1 | 20.0 |
|  | MIXED+COMPLETE | 0.3 | 20.0 |

*Table 3.* Hyper parameters.

Beyond these, all other hyperparameters are consistently applied across all benchmarks and settings. The policy, Q-function, V-function, and discriminator all utilize a 2-layer feedforward neural network architecture with 256 hidden units and ReLU activation functions. For the policy, Tanh Gaussian outputs are used. The Adam optimizer is configured with a weight decay of $1 \times 10^{-3}$, all learning rates are set to $3 \times 10^{-4}$, mini batch size is 1024, and a soft critic update parameter $\tau = 0.005$ is used. These hyperparameters are summarized in Table 4:

| Hyperparameter | Value |
|---|---|
| Network Architecture (Policy, Q-func, V-func, Discriminator) | 2-layer Neural Network |
| Hidden Units per Layer | 256 |
| Batch size | 1024 |
| Activation Function (Hidden Layers) | ReLU |
| Policy Output Activation | Tanh Gaussian |
| Optimizer | Adam |
| Learning Rate (all networks) | $3 \times 10^{-4}$ |
| Weight Decay (Adam) | $1 \times 10^{-3}$ |
| Soft Critic Update Rate ($\tau$) | 0.005 |

*Table 4.* Consistent hyperparameters used across all benchmarks and settings.

## C.5. Computational Resource

Our experiments were conducted using a pool of 12 NVIDIA GPUs, including L40, A5000, and RTX 3090 models. For each experimental configuration, five training seeds were executed in parallel, sharing a single GPU, eight CPU cores, and 64 GB of RAM. Under these shared conditions, completing 1 million training steps across all five seeds took approximately 30 minutes. The software environment was based on JAX version 0.4.28 (with CUDA 12 support), running on CUDA version 12.3.2 and cuDNN version 8.9.7.29.

Moreover, we evaluated all methods using the CHEETAH (RANDOM+EXPERT) task under identical hardware conditions:

a single NVIDIA L40 GPU, 8 CPU cores, and 64GB of RAM. For SafeDICE, we were unable to utilize GPU acceleration with TensorFlow; consequently, the method was run in CPU-only mode, resulting in slower training times. For DualCOIL, we trained two discriminators, which required approximately 5 minutes; this duration is included in the total training time. A complete training time for a single seed are reported in Table 5.

|  | DWBC-GB | SafeDICE (CPU) | SMODICE | ILID | ReCOIL | DualCOIL |
|---|---|---|---|---|---|---|
| time | ∼130 mins | ∼150 mins | ∼110 mins | ∼80 mins | ∼20 mins | ∼25 mins |

*Table 5.* Comparison of training time across methods.

# D. Additional Experiments

## D.1. Impact of the Size of the Bad Dataset: Full Details

To support the experiment in Section 6.3, we present the complete results for all MuJoCo Locomotion and Adroit manipulation tasks. In particular, we progressively increase the size of the suboptimal dataset $\mathcal{B}^B$ and evaluate the impact on each algorithm's performance. The results, shown in Figure 4, demonstrate that DualCOIL consistently outperforms all other baselines across all tasks, effectively leveraging the bad data to achieve superior performance. Notably, the results indicate that with only a single good trajectory in $\mathcal{B}^G$, increasing the number of bad trajectories in $\mathcal{B}^B$ to just 10 is sufficient for DualCOIL to achieve its highest performance across all tasks.

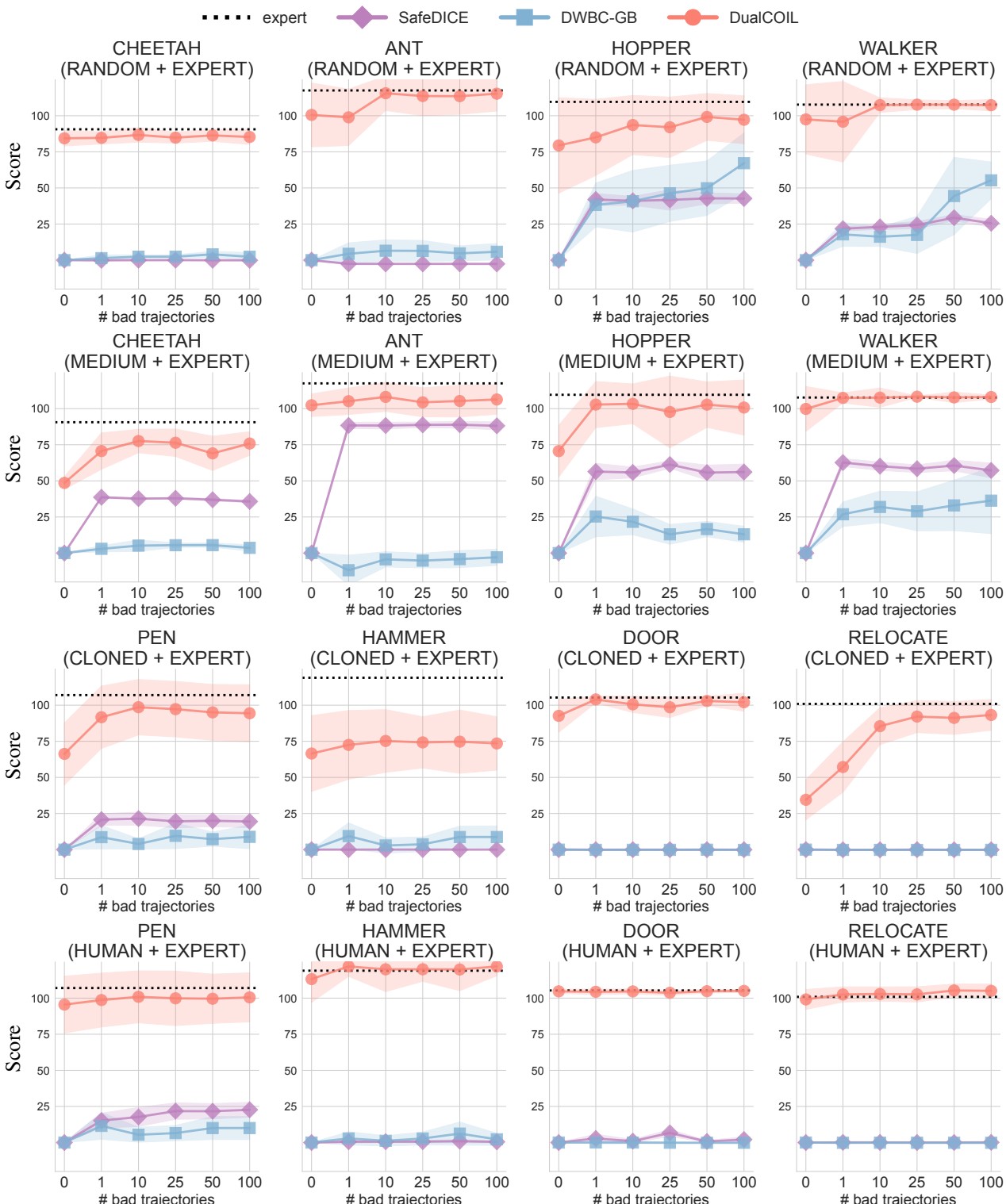

*Figure 4.* Full bad dataset size effect. SafeDICE and DWBC-GB do not have version that learn from 0 bad trajectory, we assign result 0.0 for them.

## D.2. Impact of the Number of Expert Demonstrations in good dataset $\mathcal{B}^G$

In this section, we investigate how many expert trajectories in the good dataset $\mathcal{B}^G$ are sufficient to achieve optimal performance. To this end, the quantity of expert trajectories in $\mathcal{B}^G$ was incrementally increased through the set 1,3,5,10,25, while the composition of the unlabeled dataset ($\mathcal{B}^{\text{Mix}}$) remained fixed, as specified in Table 1. The detailed results are presented in Figure 5 and 6.

ILID performs well on the Mujoco locomotion tasks (CHEETAH, ANT, HOPPER, WALKER), but struggles in 3 out of 4 Adroit tasks (HAMMER, DOOR, RELOCATE). This indicates that ILID requires a sufficient number of expert trajectories to achieve stable expert performance, which is not met in the more complex Adroit tasks. In contrast, ReCOIL appears unable to effectively leverage the good data, as its performance does not improve significantly with more expert trajectories. Overall, DualCOIL demonstrates consistently strong performance, **requiring only 3 to 5 expert trajectories** to achieve near-optimal results in all tasks.

**Discussion on the Use Cases of ILID and DualCOIL:** Through this experiment, we observe that in the Mujoco tasks, ILID can outperform DualCOIL-G when the size of the good dataset is sufficiently large. This highlights a limitation of DualCOIL, where the policy extraction objective is defined as $\max_\pi \left\{ \sum_{(s,a) \sim \mathcal{B}^U} \exp(\frac{1}{\beta} Q(s,a)) \log \pi(a|s) \right\}$. This objective uses data from the union dataset $\mathcal{B}^U$, which may assign high weights to poor-quality transitions, potentially harming training.

In contrast, ILID only retains transitions that are connected to good data and explicitly discards irrelevant or undesirable transitions (refer to the implementation details of ILID for more information). This targeted filtering strategy enables ILID to avoid the negative effects of poor transitions and scale more effectively with increasing amounts of good data.

These observations suggest a potential direction for improving DualCOIL by incorporating similar data filtering mechanisms. Specifically, enhancing DualCOIL to better isolate high-quality transitions could help it perform competitively with ILID in scenarios where the good dataset is large. We leave this exploration for future work, as it requires a careful study of how to construct an optimal dataset using Q-based methods.

In summary, ILID is a strong approach that scales well with the quality and size of the expert dataset. Practitioners may prefer discriminator-based methods like ILID when sufficient high-quality expert data is available, while DualCOIL remains a robust choice in settings where such data is limited and scalalbe with bad dataset.

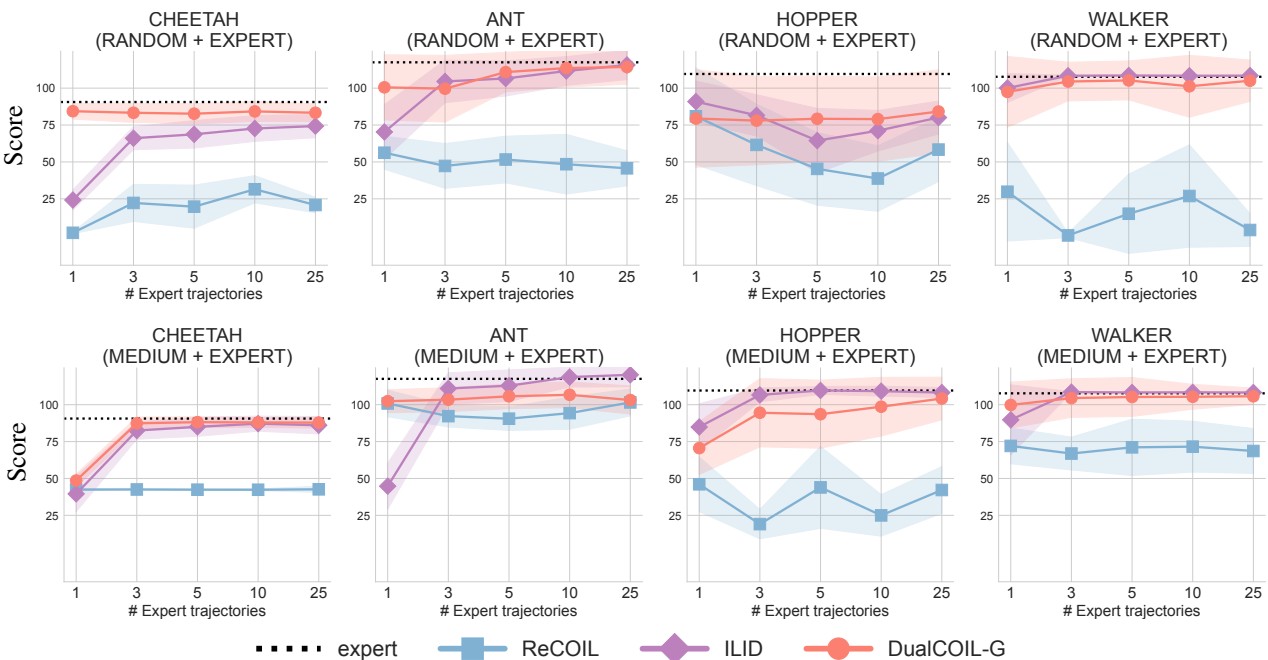

*Figure 5.* Different of good dataset size without impact from bad dataset in MuJoCo Locomotion tasks.

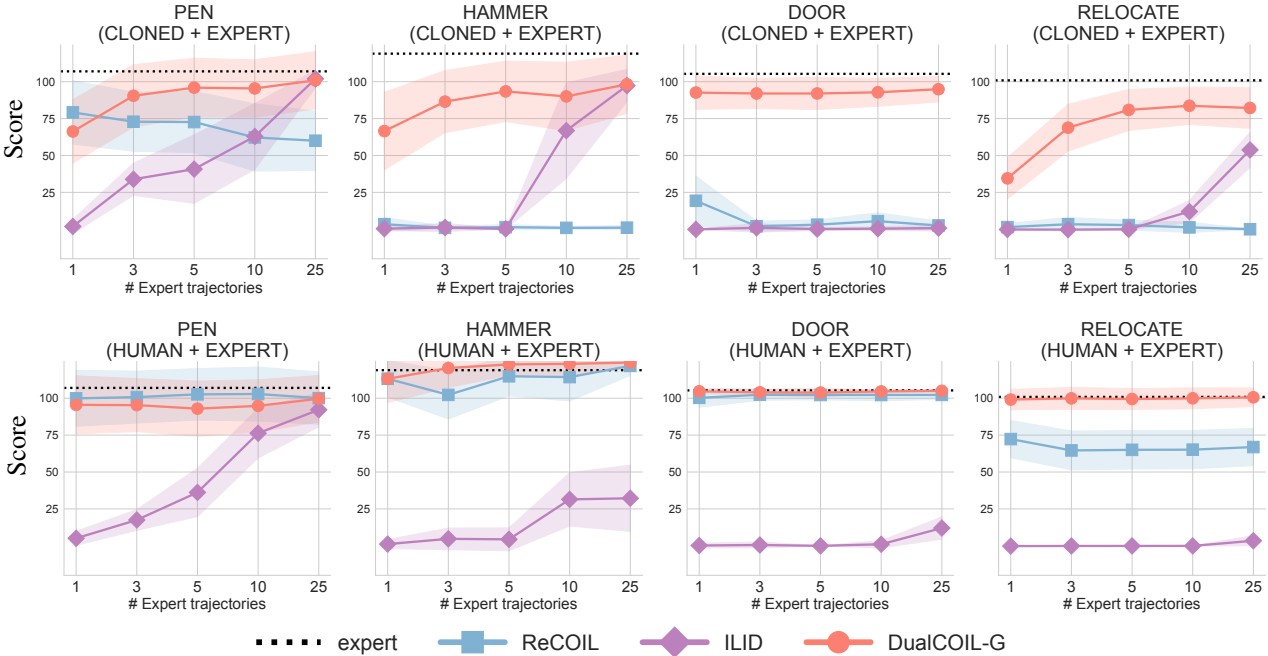

*Figure 6.* Different of good dataset size without impact from bad dataset in Adroit Manipulation tasks.

## D.3. Discussion: How Many Bad Trajectories in $\mathcal{B}^B$ Are Sufficient to Replace a Good Trajectory in $\mathcal{B}^G$ for DualCOIL?

Based on the previous experiments:

- Section D.1 addresses the question: How does the size of the bad dataset $\mathcal{B}^B$ affect the performance of DualCOIL?

- Section D.2 investigates an additional question: How does the size of the good dataset $\mathcal{B}^G$ affect the performance of DualCOIL?

From these experiments, we derive the following observations:

- With only one good trajectory in $\mathcal{B}^G$, adding 10 bad trajectories in $\mathcal{B}^B$ is sufficient for DualCOIL to achieve its best performance.

- Without any bad data $\mathcal{B}^B$, 3 to 5 good trajectories in $\mathcal{B}^G$ are enough to reach peak performance.

These results suggest that DualCOIL can efficiently utilize bad data to reduce the need for good data, with an estimated ratio of 2 to 5 bad trajectories being roughly equivalent to one good trajectory across the benchmarks studied in this paper.

## D.4. Comparison of Advantage-weighted BC and Q-weighted BC for the Policy Extraction

In this paper, we propose a novel policy extraction method called QW-BC (Objective (9)), in contrast to prior approaches that rely on AW-BC (Objective (8)). In this section, we present a comparison between QW-BC and AW-BC, as illustrated in Figure 7. Overall, QW-BC demonstrates superior policy extraction performance, attributed to its stability derived from relying on a single network estimation. In contrast, AW-BC often exhibits oscillations and instability, frequently assigning inconsistent and overly high weights to bad transitions.

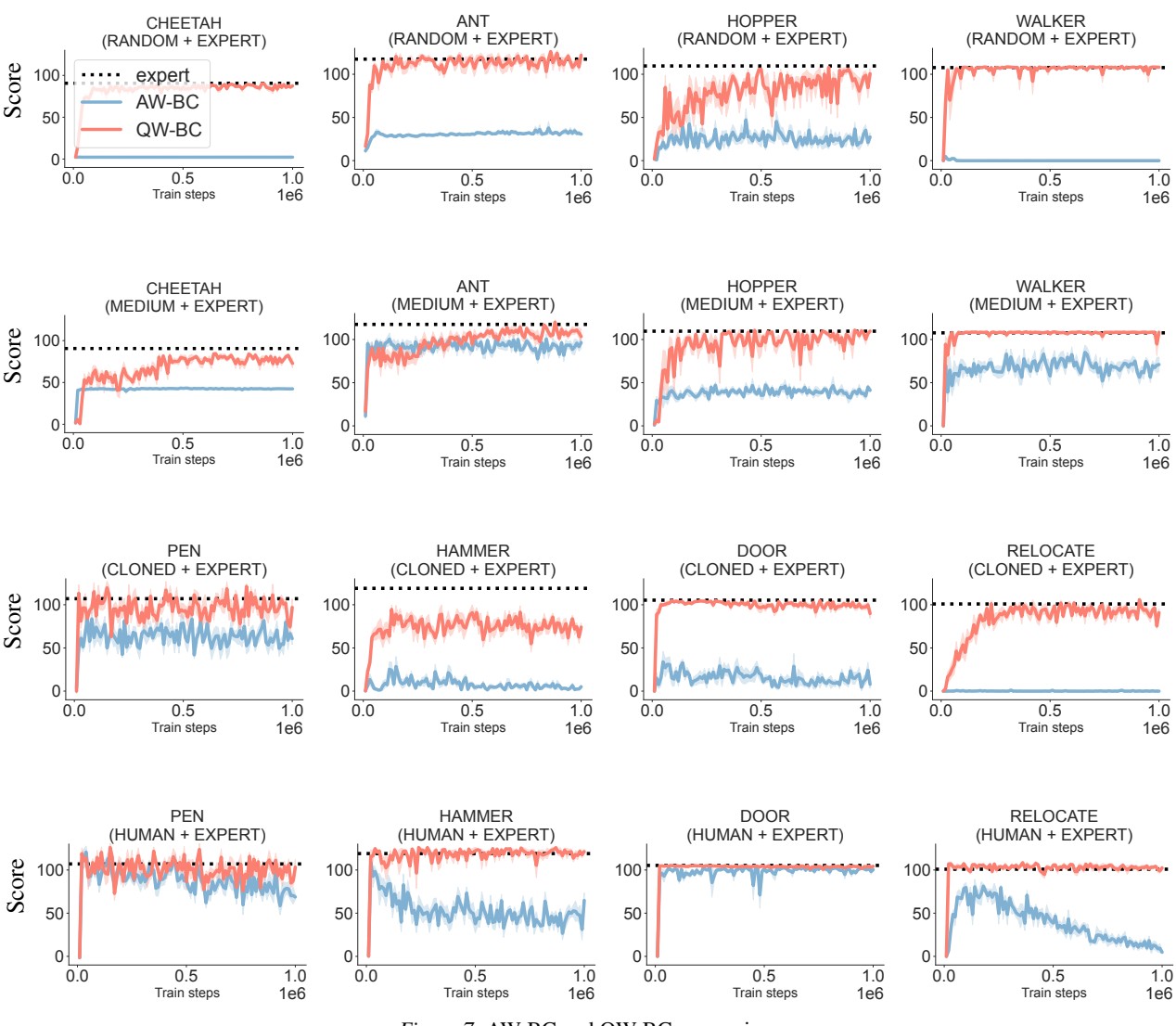

*Figure 7.* AW-BC and QW-BC comparison.

## D.5. Performance Across Varying Quality Levels of the Unlabeled Dataset $\mathcal{B}^{\text{MIX}}$

The performance of all methods is influenced by the quality of the unlabeled dataset $\mathcal{B}^{\text{MIX}}$. To evaluate the robustness of our method under varying dataset quality, we conduct experiments with different amounts of expert trajectories combined with the full set of undesirable trajectories in the unlabeled dataset. We compare our approach against ILID and ReCOIL—which leverage $\mathcal{B}^G$ and $\mathcal{B}^{\text{MIX}}$—as well as SafeDICE, which learns from $\mathcal{B}^B$ and $\mathcal{B}^{\text{MIX}}$. The detailed results of this study are presented in Figure 8.

In the Mujoco locomotion tasks, increasing the quality of the unlabeled dataset has minimal effect on SafeDICE and

ILID, and both methods continue to underperform on the Adroit hand manipulation tasks regardless of the number of expert trajectories included. In contrast, ReCOIL shows improved performance as the quality of the unlabeled dataset increases, successfully learning 4 out of 8 tasks across both locomotion and manipulation domains. Overall, our method achieves near-expert performance on 7 out of 8 tasks while requiring significantly lower-quality unlabeled datasets $\mathcal{B}^{\text{Mix}}$, demonstrating its superior data efficiency and robustness.

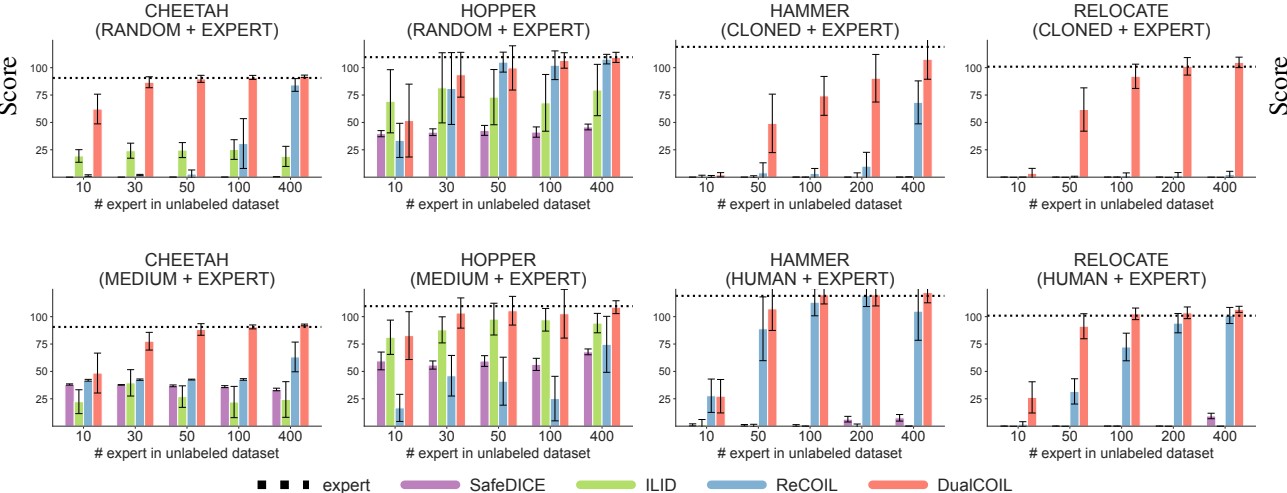

*Figure 8.* **Effect of Unlabeled Dataset Quality on Performance:** We evaluate the effect of increasing the number of expert trajectories in the unlabeled dataset $\mathcal{B}^{\text{Mix}}$. The results are calculated from 5 different training seeds, reported in normalized score. Our method outperforms SafeDICE, ILID and ReCOIL across both locomotion and manipulation tasks, achieving near-expert performance on most environments even with a small number of expert demonstrations.

## D.6. Effect of Increasing the Proportion of Bad Data in the Unlabeled Dataset

In this experiment, we maintain the same good dataset ($\mathcal{B}^G$) and bad dataset ($\mathcal{B}^B$) as used in the main comparison in Section 6.2. Our modification focuses on the unlabeled dataset ($\mathcal{B}^{MIX}$). Within $\mathcal{B}^{MIX}$, the number of EXPERT trajectories remains consistent with Section 6.2, but the RANDOM dataset within it is duplicated multiple times to increase the proportion of bad data (each dataset contain about 1000 RANDOM trajectories). The results, presented in Table 6, indicate that increasing the amount of poor-quality data leads to a general decline in performance across all methods. Nevertheless, our algorithm remains consistently robust and continues to outperform the main baselines under these more challenging conditions.

## D.7. Experiments with Extremely Large Bad Dataset

Although we previously examined the effect of the size of the bad dataset $\mathcal{B}^B$ in Appendix D.1, that study was restricted to at most 100 trajectories. In this experiment, we aim to further investigate how enlarging $\mathcal{B}^B$ can improve performance. Since the RANDOM dataset from D4RL is relatively small (which limit the analysis of Appendix D.1), we augment it by generating additional random trajectories through direct interaction with the environment. The experiment results are shown in Table 7.

From the results, we observe that increasing the quantity of bad demonstrations generally leads to improved performance in most cases. This is likely because a richer set of bad data provides better coverage of the undesirable regions in the action space, which helps the algorithm more effectively learn what to avoid. However, in a few cases, the performance either improves only marginally or even slightly decreases. This can be attributed to the fact that adding more bad demonstrations does not always guarantee broader or more informative coverage of poor actions. If the additional bad data is redundant or fails to introduce new undesirable behavior patterns, its benefit to learning may be limited or even slightly detrimental due to noise.

| CHEETAH (RANDOM + EXPERT) | 1xRANDOM | 2xRANDOM | 3xRANDOM | 5xRANDOM |
|---|---|---|---|---|
| SafeDICE | $-0.0_{\pm 0.0}$ | $-0.0_{\pm 0.0}$ | $-0.0_{\pm 0.0}$ | $-0.0_{\pm 0.1}$ |
| ILID | $21.1_{\pm 7.6}$ | $29.0_{\pm 1.4}$ | $24.7_{\pm 4.0}$ | $26.7_{\pm 0.4}$ |
| ReCOIL | $2.0_{\pm 0.6}$ | $2.3_{\pm 0.1}$ | $2.0_{\pm 0.2}$ | $1.8_{\pm 0.7}$ |
| DualCOIL | $\mathbf{86.7}_{\pm 5.0}$ | $\mathbf{81.8}_{\pm 2.7}$ | $\mathbf{75.9}_{\pm 2.1}$ | $\mathbf{59.9}_{\pm 2.5}$ |
| RELOCATE (CLONED + EXPERT) | 1xRANDOM | 2xRANDOM | 3xRANDOM | 5xRANDOM |
| SafeDICE | $-0.1_{\pm 0.0}$ | $-0.1_{\pm 0.0}$ | $-0.1_{\pm 0.0}$ | $-0.1_{\pm 0.0}$ |
| ILID | $-0.1_{\pm 0.1}$ | $-0.2_{\pm 0.1}$ | $-0.2_{\pm 0.0}$ | $-0.2_{\pm 0.0}$ |
| ReCOIL | $1.4_{\pm 2.4}$ | $0.4_{\pm 0.3}$ | $0.1_{\pm 0.0}$ | $0.1_{\pm 0.1}$ |
| DualCOIL | $\mathbf{92.1}_{\pm 11.1}$ | $\mathbf{64.7}_{\pm 2.4}$ | $\mathbf{35.8}_{\pm 14.3}$ | $\mathbf{9.3}_{\pm 9.2}$ |
| KITCHEN (PARTIAL + COMPLETE) | 1xRANDOM | 2xRANDOM | 3xRANDOM | 5xRANDOM |
| SafeDICE | $2.8_{\pm 1.1}$ | $3.8_{\pm 2.3}$ | $4.9_{\pm 1.5}$ | $3.0_{\pm 1.2}$ |
| ILID | $27.3_{\pm 5.4}$ | $7.6_{\pm 9.7}$ | $13.0_{\pm 4.9}$ | $11.3_{\pm 4.4}$ |
| ReCOIL | $48.8_{\pm 8.3}$ | $41.6_{\pm 1.8}$ | $44.5_{\pm 3.7}$ | $44.3_{\pm 8.2}$ |
| DualCOIL | $\mathbf{53.1}_{\pm 13.1}$ | $\mathbf{57.6}_{\pm 5.4}$ | $\mathbf{56.5}_{\pm 9.2}$ | $\mathbf{56.8}_{\pm 7.0}$ |

*Table 6.* Increase the proportion of bad in the unlabeled dataset $\mathcal{B}^{MIX}$ in three different environments.

| CHEETAH (RANDOM + EXPERT) | 100 | 300 | 500 | 1000 |
|---|---|---|---|---|
| DWBC-GB | $2.3_{\pm 2.9}$ | $1.4_{\pm 1.3}$ | $3.0_{\pm 2.3}$ | $3.2_{\pm 2.1}$ |
| SafeDICE | $-0.1_{\pm 0.1}$ | $0.0_{\pm 0.0}$ | $0.3_{\pm 0.1}$ | $0.5_{\pm 0.3}$ |
| DualCOIL | $\mathbf{85.3}_{\pm 5.1}$ | $\mathbf{91.4}_{\pm 1.5}$ | $\mathbf{91.8}_{\pm 1.0}$ | $\mathbf{91.5}_{\pm 0.8}$ |
| RELOCATE (CLONED + EXPERT) | 100 | 300 | 500 | 1000 |
| DWBC-GB | $-0.1_{\pm 0.1}$ | $-0.2_{\pm 0.0}$ | $-0.2_{\pm 0.0}$ | $-0.2_{\pm 0.0}$ |
| SafeDICE | $-0.1_{\pm 0.0}$ | $-0.1_{\pm 0.0}$ | $-0.1_{\pm 0.0}$ | $-0.1_{\pm 0.0}$ |
| DualCOIL | $\mathbf{93.2}_{\pm 10.7}$ | $\mathbf{96.1}_{\pm 12.0}$ | $\mathbf{96.2}_{\pm 11.2}$ | $\mathbf{97.9}_{\pm 11.6}$ |
| KITCHEN (PARTIAL + COMPLETE) | 100 | 300 | 500 | 1000 |
| DWBC-GB | $17.0_{\pm 9.8}$ | $14.8_{\pm 6.7}$ | $10.5_{\pm 9.2}$ | $15.3_{\pm 8.9}$ |
| SafeDICE | $2.7_{\pm 2.6}$ | $1.7_{\pm 0.7}$ | $1.9_{\pm 1.4}$ | $0.4_{\pm 0.2}$ |
| DualCOIL | $\mathbf{60.3}_{\pm 10.6}$ | $\mathbf{63.8}_{\pm 9.2}$ | $\mathbf{59.2}_{\pm 8.7}$ | $\mathbf{60.8}_{\pm 9.4}$ |

*Table 7.* Increasing size of Bad dataset $\mathcal{B}^{B}$.

## D.8. Comparison with Adapted Offline RL Methods

In this section, we compare our approach with offline RL methods adapted to learn from both good and bad datasets by assigning rewards of $+1$ to $\mathcal{B}^G$ and $-1$ to $\mathcal{B}^B$, and combining all three datasets into a single offline training set. We evaluate against two widely used baselines, CQL (Kumar et al., 2020) and IQL (Kostrikov et al., 2021), using the same dataset sizes as in Section 6.2 for fairness. The results in Table 8 show that our method consistently outperforms both baselines.

|          | CHEETAH | HOPPER | HAMMER | RELOCATE | KITCHEN |
|----------|---------|--------|--------|----------|---------|
| CQL      | $-2.3_{\pm 1.1}$ | $26.8_{\pm 13.6}$ | $0.3_{\pm 0.0}$ | $-0.3_{\pm 0.0}$ | $0.0_{\pm 0.0}$ |
| IQL      | $-0.5_{\pm 0.6}$ | $4.6_{\pm 2.8}$ | $4.4_{\pm 3.5}$ | $-0.1_{\pm 0.0}$ | $11.5_{\pm 6.5}$ |
| DualCOIL | $\mathbf{86.7}_{\pm 5.0}$ | $\mathbf{93.6}_{\pm 20.5}$ | $\mathbf{74.3}_{\pm 17.8}$ | $\mathbf{92.1}_{\pm 11.1}$ | $\mathbf{53.1}_{\pm 13.1}$ |

*Table 8.* Comparison of DualCOIL with offline RL methods.

## D.9. Discussion: Distribution-matching Approach vs Preference-based Approach

The good and bad data setup is reminiscent of preference-based methods. In this section, we want to discuss the difference between our approach (distribution-matching) and preference-based approach with two keys aspects:

- **Input data construction:** Our approach is based on contrastive demonstrations, explicitly labeled as good or bad. In contrast, preference-based methods rely on pairwise preference feedback between trajectories, where both trajectories can be good, bad, or of similar quality.

- **Learning objective:** ConstraDICE is designed to **explicitly imitate expert behavior while avoiding bad behavior**. Preference-based methods, on the other hand, aim to infer a reward function or policy that aligns with the provided preferences, without necessarily distinguishing between good and bad demonstrations in an absolute sense.

Intuitively, this means preference-based learning is conceptually different and not well-suited to our setting. Simply enforcing a preference like $r(good) > r(bad)$ does not capture the critical requirement of explicitly avoiding bad behaviors. Even if the method assigns lower rewards to bad trajectories, it does not guarantee that the resulting policy will avoid them.

To empirically support this argument, we conducted additional experiments using an offline preference-based learning approach which is IPL (Hejna & Sadigh, 2024) with the configuration $r(good) > r(bad)$. The results, presented in Table 9, further demonstrate that preference-based methods fail to learn effective policies in our contrastive good-bad setting.

|          | CHEETAH | HOPPER | HAMMER | RELOCATE | KITCHEN |
|----------|---------|--------|--------|----------|---------|
| IPL      | $1.5_{\pm 0.1}$ | $6.4_{\pm 0.4}$ | $0.5_{\pm 0.1}$ | $-0.1_{\pm 0.0}$ | $34.7_{\pm 3.7}$ |
| DualCOIL | $\mathbf{86.7}_{\pm 5.0}$ | $\mathbf{93.6}_{\pm 20.5}$ | $\mathbf{74.3}_{\pm 17.8}$ | $\mathbf{92.1}_{\pm 11.1}$ | $\mathbf{53.1}_{\pm 13.1}$ |

*Table 9.* Comparison of DualCOIL with IPL.

## D.10. Comparison with UNIQ: A State-of-the-Art Algorithm for Learning from Bad Demonstrations

In this section, we present an additional experiment comparing our approach with UNIQ (Hoang et al., 2024b), a state-of-the-art method specifically designed to avoid bad demonstrations (similar to SafeDICE). While both UNIQ and DUALCOIL address offline imitation learning under the presence of bad-quality demonstrations, the two methods are fundamentally different in both formulation and learning principle. Specifically, UNIQ builds upon the IQ-Learn framework (Garg et al., 2021), which optimizes a max–min objective over reward and policy using an entropy-regularized formulation. In contrast, DUALCOIL is derived from the DUARL and DICE frameworks (Sikchi et al., 2024), employing a tractable reformulation based on minimizing the KL divergence between state–action visitation distributions. This yields a Q-learning–style objective that is computationally simpler and more stable to optimize. Moreover, DUALCOIL introduces a surrogate

approximation (Section 4.2) that further enhances efficiency without sacrificing alignment with the original theoretical objective.

For consistency, we adopt the same dataset setup as in Section 6.2, where learning is performed using $\mathcal{B}^B$ only. The results in Figure 9 show that, with expert support, DualCOIL achieves the best overall performance.

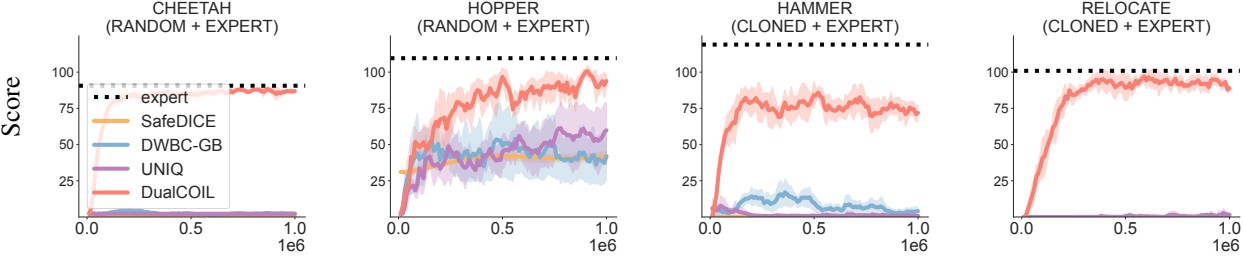

*Figure 9.* Comparison with UNIQ.

## D.11. Adaptations and Experiments with $\alpha > 1$

From our objective function (1), we introduce a hyperparameter $0 \leq \alpha < 1$, which controls the weighting of the bad data objective—this corresponds to question **(Q3)**. To evaluate the sensitivity of our method to $\alpha$, we conduct experiments by varying its value and observing its impact on final performance. Specifically, we perform a full sweep over $\alpha \in \{0, 0.1, 0.2, \ldots, 0.9\}$ to illustrate how this key hyperparameter influences learning outcomes.

Interestingly, we observe that in some cases, settings with $\alpha \geq 1$ yield favorable performance, suggesting that avoiding bad data may, at times, be more critical than imitating good data. However, directly applying $\alpha \geq 1$ in our original formulation violates convexity conditions.

To address this, we propose a naive modification of Objective (7) that accommodates $\alpha \geq 1$ while preserving practical applicability. The revised objective is defined as:

$$\widetilde{L}(Q \mid V) = (1 - \gamma) \, \mathbb{E}_{s \sim p_0} \left[ V(s) \right] - \mathbb{E}_{(s,a) \sim d^U} \left[ \exp \left( \Psi(s,a) \right) \left( Q(s,a) - \gamma \mathbb{E}_{s'} \left[ V(s') \right] \right) \right], \tag{21}$$

which enables empirical investigation into the high-$\alpha$ regime while sidestepping theoretical limitations. The experiment results are provided in Figure 10. Overall, $\alpha \geq 1$ does not provide good performance, which raises the limitation of the naive adaptation.

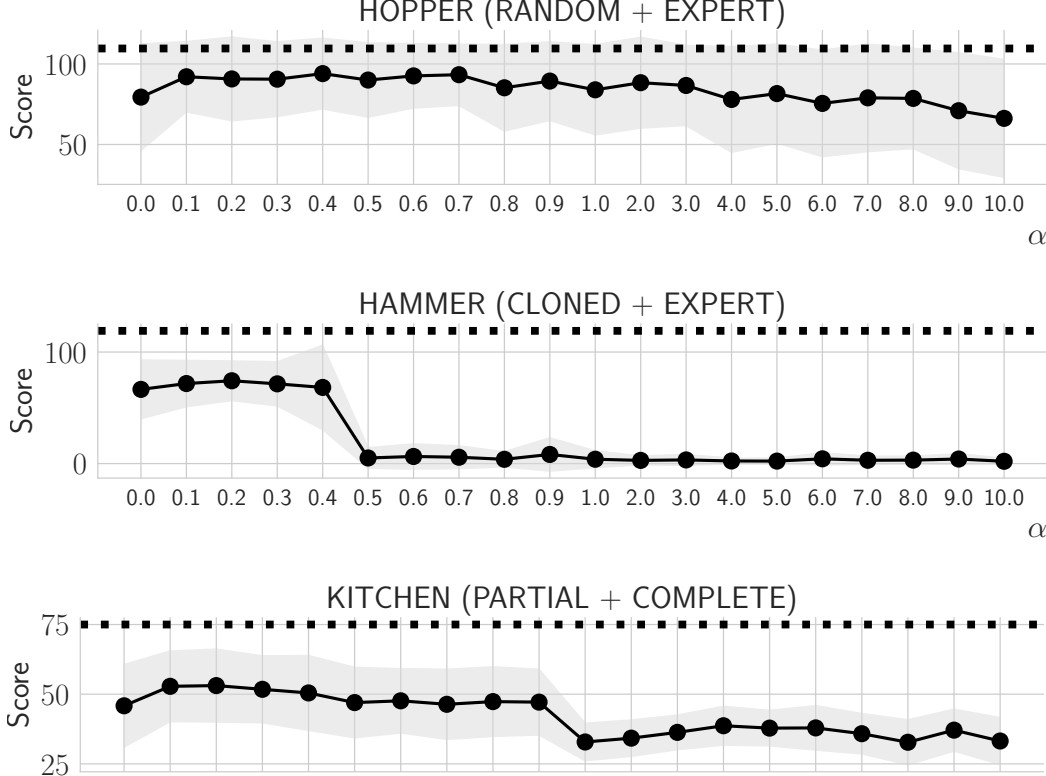

*Figure 10.* Performance of large $\alpha \geq 1$.

## D.12. Comparison Between $L(Q, \pi)$ and the Surrogate $\widetilde{L}(Q, \pi)$

As shown in Proposition 4.3, the original objective $L(Q \mid V)$ ((4)) is transformed into a modified version $\widetilde{L}(Q \mid V)$ ((7)). This experiment investigates the performance differences between the two objectives.z

To improve the stability of the original objective $L(Q \mid V)$, we need to address the issue of exponential terms producing extremely large values, which can lead to numerical instability. A practical approach is to clip the input to the exponential function to a bounded range $[\mathrm{minR}, \mathrm{maxR}]$, resulting in the following formulation:

$$
\begin{aligned}
L(Q, \pi) =& (1 - \gamma) \, \mathbb{E}_{s \sim p_0} \left[ V_Q^\pi(s) \right] \\
&+ (1 - \alpha) \mathbb{E}_{(s,a) \sim d^U} \left[ \exp \left( \left( \frac{\Psi(s,a) - \mathcal{T}^\pi[Q](s,a)}{1 - \alpha} \right) . \mathrm{clip}(\mathrm{minR}, \mathrm{maxR}) \right) \right],
\end{aligned}
\tag{22}
$$

where $\mathrm{minR} = -7$ and $\mathrm{maxR} = 7$ in our experiments.

The results of this ablation study are presented in Figure 11, illustrating the performance impact of this stability-enhancing modification. In general, the clipping technique effectively mitigates the instability caused by the exponential term, successfully preventing $NaN$ errors during training. However, this modification also leads to a drop in performance and, in some tasks, causes the method to fail to learn effectively.

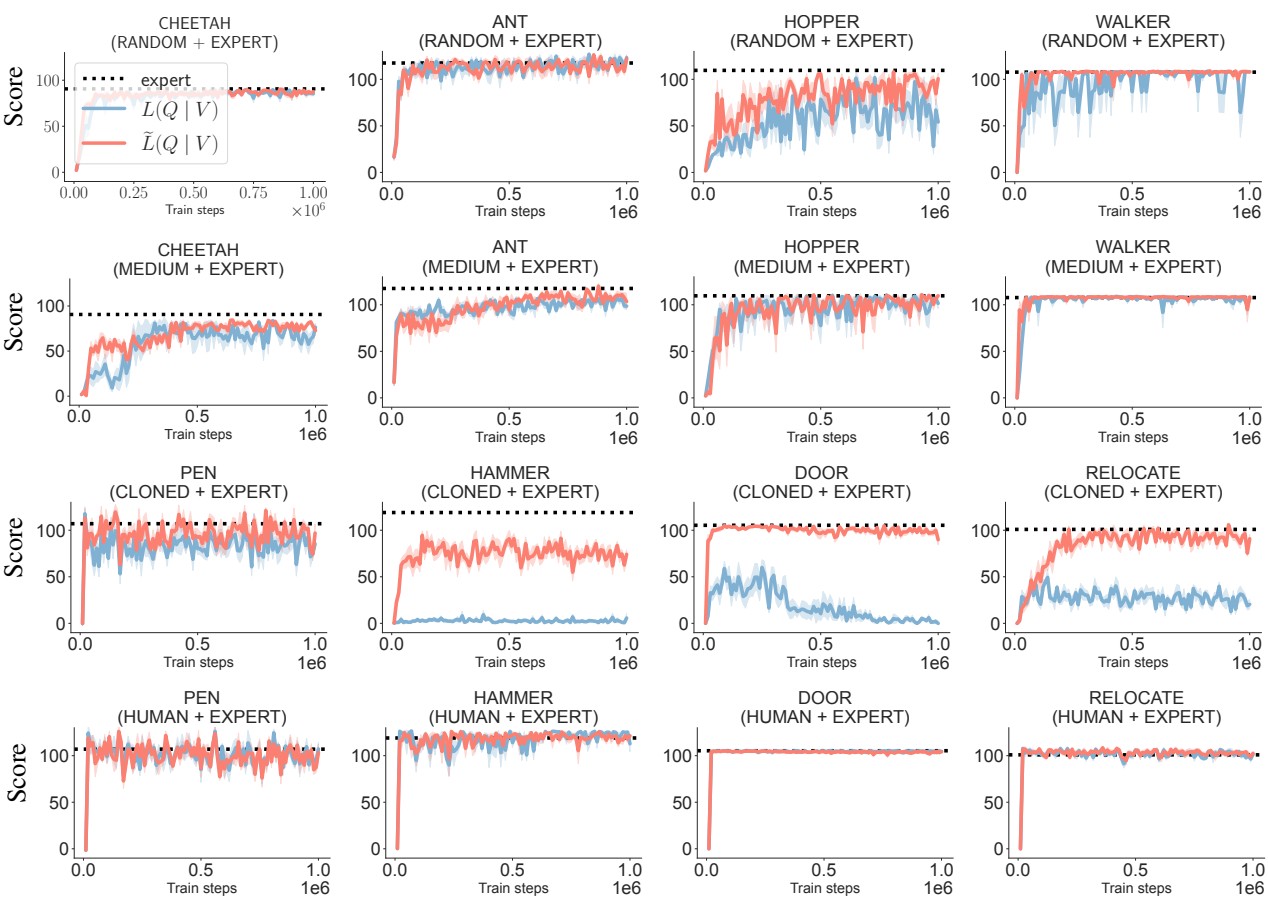

*Figure 11.* Exponetial ablation study.

## D.13. Sensitivity Analysis of $\beta$

In this section, we explore how different values of the $\beta$ parameter affect performance. The experiment results are provided in Table 10. The results show that while $\beta$ significantly influences outcomes, performance remains consistent over a wide range of $\beta$ values, implying that minimal tuning effort is needed for this hyperparameter.

| Task | unlabeled $\mathcal{B}^{\text{Mix}}$ | $\beta$ value | | | | | | |
|---|---|---|---|---|---|---|---|---|
| | | 1 | 3 | 5 | 10 | 15 | 20 | 30 |
| CHEETAH | RANDOM+EXPERT | $2.25_{\pm 0.0}$ | $2.25_{\pm 0.0}$ | $2.25_{\pm 0.0}$ | $2.24_{\pm 0.0}$ | $83.2_{\pm 5.3}$ | $\mathbf{85.8}_{\pm 2.1}$ | $84.3_{\pm 1.4}$ |
| | MEDIUM+EXPERT | $42.4_{\pm 0.2}$ | $42.9_{\pm 0.3}$ | $53.9_{\pm 8.8}$ | $\mathbf{83.1}_{\pm 4.9}$ | $80.1_{\pm 2.6}$ | $78.7_{\pm 2.3}$ | $76.7_{\pm 5.2}$ |
| ANT | RANDOM+EXPERT | $39.5_{\pm 7.3}$ | $69.3_{\pm 6.5}$ | $60.9_{\pm 28.7}$ | $115.6_{\pm 4.6}$ | $\mathbf{118.0}_{\pm 2.1}$ | $114.5_{\pm 1.7}$ | $116.0_{\pm 2.1}$ |
| | MEDIUM+EXPERT | $91.0_{\pm 1.1}$ | $90.6_{\pm 1.7}$ | $93.7_{\pm 1.5}$ | $104.8_{\pm 3.9}$ | $\mathbf{106.5}_{\pm 2.4}$ | $101.1_{\pm 3.3}$ | $95.1_{\pm 1.3}$ |
| HOPPER | RANDOM+EXPERT | $4.7_{\pm 0.4}$ | $5.2_{\pm 0.9}$ | $7.2_{\pm 1.3}$ | $7.9_{\pm 1.9}$ | $20.4_{\pm 9.7}$ | $67.4_{\pm 7.9}$ | $\mathbf{94.4}_{\pm 6.3}$ |
| | MEDIUM+EXPERT | $52.1_{\pm 1.5}$ | $46.0_{\pm 1.0}$ | $85.8_{\pm 11.6}$ | $96.3_{\pm 8.1}$ | $96.9_{\pm 12.5}$ | $\mathbf{99.6}_{\pm 4.1}$ | $98.0_{\pm 5.7}$ |
| WALKER | RANDOM+EXPERT | $2.9_{\pm 2.6}$ | $3.5_{\pm 2.9}$ | $6.4_{\pm 4.6}$ | $32.5_{\pm 27.7}$ | $105.7_{\pm 4.5}$ | $106.2_{\pm 2.0}$ | $\mathbf{107.5}_{\pm 1.1}$ |
| | MEDIUM+EXPERT | $68.3_{\pm 3.7}$ | $65.8_{\pm 3.2}$ | $53.4_{\pm 3.6}$ | $104.9_{\pm 2.5}$ | $108.1_{\pm 0.1}$ | $\mathbf{108.2}_{\pm 0.2}$ | $\mathbf{108.2}_{\pm 0.1}$ |

*Table 10.* Performance of DualCOIL in different $\beta$ value in MuJoCo locomotion tasks.

## D.14. Extreme-V versus Log-sum-exp for Value Function Estimation

In Section 5, we use Extreme-V to estimate the value function $V$ instead of LogSumExp to handle continuous action spaces. However, in prior work related to soft Q-learning, in the continuous action space, researchers estimate $V$ based on the current $Q$ and policy as $V(s) = Q(s, \pi(a|s)) + \alpha H(\pi)$, where $H(\pi)$ is the entropy of the policy $\pi$ and $\alpha$ is a multiplier controlling the contribution of the entropy to the value function. In this section, we compare the performance of Extreme-V with this entropy-based estimation. We follow the same $\alpha$ as LS-IQ (Al-Hafez et al., 2023) for MuJoCo tasks, while setting a fixed $\alpha = 0.01$ for all Adroit tasks. The comparison results are provided in Figure 12.

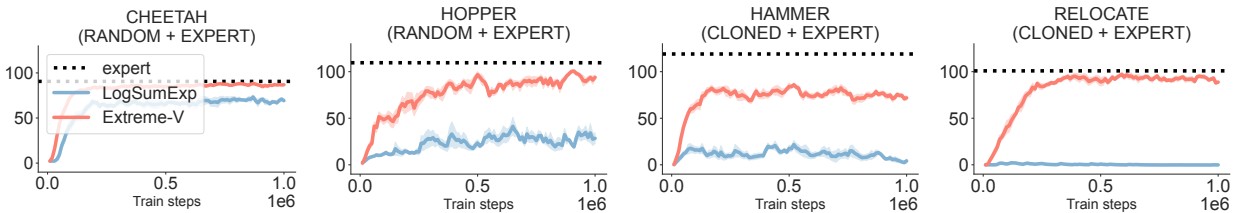

*Figure 12.* Compare Extreme-V with LogSumExp.

## D.15. DualCOIL for Safety Tasks

In this section, we evaluate our method in a safe domain where the objective is to maximize return while ensuring the cost remains below a specified threshold. We utilize the Safety Gymnasium benchmark for this experiment, specifically the SafetyPointGoal1 and SafetyPointButton1 environments, with a cost threshold of 25.0. Trajectories with an accumulated cost exceeding 25.0 are considered unsafe, while those with an accumulated reward below a specific return threshold are categorized as low-return.

We generate three types of policies: safe high-return (using PPO-Lag), unsafe high-return (using PPO), and low-return (using a random policy). The detailed quality of these datasets is reported in Table 11. Using these policies, we construct the following datasets:

- Good dataset ($D^G$): Consists of safe high-return trajectories.

- Bad dataset ($D^B$): Consists of unsafe high-return and low-return trajectories.

| Policy type | SafetyPointGoal1 | | SafetyPointButton1 | |
|---|---|---|---|---|
| | Return | Cost | Return | Cost |
| Safe High-Return | $24.4_{\pm 2.6}$ | $11.5_{\pm 19.0}$ | $10.2_{\pm 5.0}$ | $16.4_{\pm 33.3}$ |
| Unsafe High-Return | $24.2_{\pm 1.6}$ | $59.8_{\pm 41.3}$ | $10.2_{\pm 3.2}$ | $135.3_{\pm 37.2}$ |
| Low-Return | $-4.0_{\pm 1.6}$ | $21.8_{\pm 43.9}$ | $-0.3_{\pm 0.9}$ | $42.7_{\pm 79.2}$ |

*Table 11.* Quality of datasets in two tasks SafetyPointGoal1 and SafetyPointButton1 in mean and standard deviation.

- Unlabeled dataset ($D^{MIX}$): A mixture of unsafe high-return, safe high-return, and low-return trajectories.

In our experiments, we fix the size of $D^G$ to 5 safe high-return trajectories. $D^{MIX}$ is composed of 500 unsafe high-return, 500 low-return, and 100 safe high-return trajectories. We aim to verify DualCOIL's ability to leverage varying sizes of $D^B$ (comprising $50\%$ unsafe high-return and $50\%$ low-return data) to assist training, thereby helping the policy avoid constraint violations and low-return behaviors.

For evaluation, we test the policy over 100 independent episodes. We compute the violation rate, defined as the proportion of episodes where the return falls below the specified return threshold and the cost exceeds the cost threshold. Additionally, we report the average Return and Cost for these runs. Detailed results are provided in Figure 13. Overall, there are a clearly trend that increasing the size of bad dataset $D^B$ lead to lower violation rate.

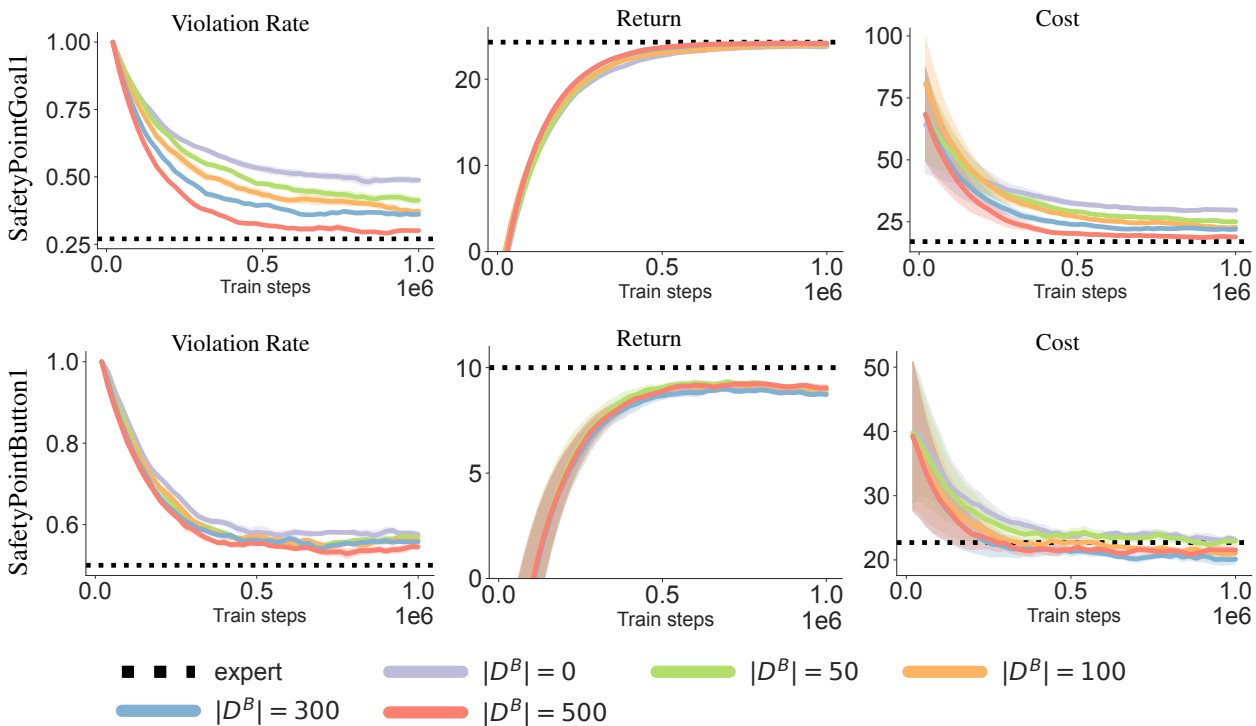

*Figure 13.* Performance of DualCOIL across different sizes of the bad dataset $D^B$ on Safety Gymnasium tasks. The violation rate reflects the proportion of unsafe or low-return behaviors exhibited by the current policy during evaluation (lower is better). Higher return indicates better task performance, while lower cost corresponds to safer behavior.

