# OpenReview forum: "DualCOIL: Offline Imitation Learning from Contrasting Demonstrations"
_ICML.cc/2026/Conference — ICML 2026 regular_

### Official Review · Reviewer_EEK2 · 2026-02-26

**Soundness:** 3
**Presentation:** 3
**Significance:** 3
**Originality:** 3
**Overall Recommendation:** 5
**Confidence:** 3

**Summary:**

This work introduces DualCOIL, an offline imitation / reinforcement learning method with a Dual-KL optimization objective.

The KL divergence is used to steer the learned model towards behavior from a known expert demonstration dataset and away from a known bad demonstration dataset. Extensive experiments on D4RL benchmarks show the performance increase from this approach and ablate over different settings, such as varying number of labeled demonstrations.

**Compliance With Llm Reviewing Policy:**

Affirmed.

**Final Justification:**

The rebuttal helped to clarify some open questions and I raised my score accordingly.

**Key Questions For Authors:**

1. How does this method perform with a varying number of good demonstrations?
2. Sec 6.5 evaluates the robustness towards the hyperparameter alpha. However, in the experiment Hammer model performance strongly degrades. Do the author's have an intuition why this might be the case for this experimental setting?

**Limitations:**

yes

**Strengths And Weaknesses:**

## Strengths
- clear presentation
- elegant method with use of Dual-KL objective
- very reasonable and mild assumptions with a small number of identified expert demonstrations, majority unlabeled demonstrations and a experimentally shown robustness to required number of identified bad demonstrations. Authors are open about their limitation that this method does not prevent the learned model to still perform actions from the bad demonstration set and should not be considered a safety constraint.
- thorough experimental evaluation and ablation

## Weaknesses
- I recommend an additional ablation with regard to the number of good demonstrations, similar to the number of bad demonstrations. In my understanding, the evaluating the number of bad demonstrations potentially reduces the labeling effort. Reducing the number of required expert demonstrations could further strengthen this contribution.

---

> ### Author Rebuttal · Authors · 2026-03-30
>
> We thank the reviewer for carefully reading our paper and providing insightful feedback. Please find below our detailed responses to each comment.
>
> ---
> > I recommend an additional ablation with regard to the number of good demonstrations, similar to the number of bad demonstrations. In my understanding, the evaluating the number of bad demonstrations potentially reduces the labeling effort. Reducing the number of required expert demonstrations could further strengthen this contribution.
>
> Thank you for your comment. We would like to clarify that, in our experiments, Our setting already considers the extreme low-data regime (1 expert trajectory). Moreover, we provide an additional experiment in Appendix D.2 where we increase the number of expert demonstrations and ignore the impact of bad demonstrations to better understand the effect of expert data on our method.
>
> > How does this method perform with a varying number of good demonstrations?
>
> Thank you for your comment. We already have an ablation study in Appendix D.2 with varying numbers of good demonstrations.
>
> > Sec 6.5 evaluates the robustness towards the hyperparameter alpha. However, in the experiment Hammer model performance strongly degrades. Do the author's have an intuition why this might be the case for this experimental setting?
>
> We thank the reviewer for highlighting this. The performance drop in the HAMMER task at high $\alpha$ stems from strong overlap between the “good” (EXPERT) and “bad” (CLONED) datasets, which makes the discriminator produce noisy and indistinguishable signals.
>
> As a result, increasing $\alpha$ suppresses the true reward signal—by miss classifying expert states as bad—and amplifies this error through the exponential term. This leads to unstable and degraded learning. Empirically, relying primarily on good data with only a small penalty on bad data yields the best performance.
>
> ---
> We hope that the above responses address your questions and concerns. Should you have any further inquiries, we would be happy to consider and respond to them.

---

> > ### Author Rebuttal · Reviewer_EEK2 · 2026-04-01
> >
> > I thank the authors for their thorough responses to all reviews and highlighting the experiments included in the Appendix.
> > I will raise my score accordingly.
> >
> >
> > Lastly, I want to highlight the new experiment performed in response to Reviewer 1 (S26o) regarding mislabeled data.
> > I believe this to be an important analysis and strongly recommend adding this experiment to the main body of the manuscript to  aid future readers in adopting this approach. The current appendix is already very thorough (great!), but this also means it is easy to overlook important insights (see myself).

---

### Official Review · Reviewer_Nta9 · 2026-03-10

**Soundness:** 3
**Presentation:** 3
**Significance:** 2
**Originality:** 1
**Overall Recommendation:** 3
**Confidence:** 4

**Summary:**

The paper proposes a method for offline imitation learning that also accounts for negative examples (demonstrations that should be avoided). This problem is framed as minimizing the KL divergence to the good demonstrations while maximizing the KL divergence to the negative demonstrations. The main derivations are similar to related works (e.g. ReCOIL), but differ by introducing a lower bound optimization and by using the value function instead of the advantage function for policy extraction. The resulting method achieves good performance on typical offline IL benchmarks, even when not using negative examples.

**Compliance With Llm Reviewing Policy:**

Affirmed.

**Final Justification:**

After considering the rebuttal and the other reviews, I still believe that the paper is borderline and below the acceptance threshold.

- Large parts of the main method section are direct application of prior results (e.g. by Sikchi et al.) without proper attribution. This makes it difficult to disentangle the contributions of the current work from prior work.

- The key contribution (as stated by the authors) is approximating the loss by a lower bound, which improves performance in practice. However, this heuristic lacks a proper theoretical justifications. The rebuttal only provided the hand-wavy arguments that the approximation "preserves the optimization structure". The follow-up reply argued that replacing an exponential term by a linear term is justified because both increase with x, which is also not convincing, since it can still greatly effect the optimum of the overall objective.

Overall, these issues overweight the merits of empirically improved performance in a rather specific problem setting of offline imitation learning with explicitly bad demonstrations.

**Key Questions For Authors:**

My main concerns are related to the originality, and the theoretical justification of the lower bound and the KL maximization.

On top of addressing these issues, I would like the authors to relate the lower bound with semi-gradients, which are commonly used to get rid of the exponential (see "AN OPTIMAL DISCRIMINATOR WEIGHTED IMITATION PERSPECTIVE FOR REINFORCEMENT LEARNING").

**Limitations:**

yes

**Strengths And Weaknesses:**

## Soundness

### Strengths
1. __Claims are supported by derivations.__ The claims that the objective is still convex for $\alpha < 1$, that the surrogate objective is a lower bound and can non-adversarial, and that AWR can be performed with the Q function instead of the advantage function are supported by derivations.
2. __Good empirical performance.__ The method achieves good performance on typical benchmarks, in particular when including negative demonstrations.
3. __Code submitted.__ The supplementary includes the source code, which helps reproducibility.

### Weaknesses
1. __Optimization problem is not well motivated.__ The proposed method maximizes a reverse KL between the distributions of the agent and bad demonstratons. The reverse KL is mode-seeking, which can be sensible when minimizing it, but not well-suited for maximization. If we solely focus on the objective of maximizing the reverse KL to the bad distribution, we can see that the optimal distribution would collapse to that is fathest away from the bad demonstrations, which is likely not what we want. Instead of collapsing to a region where the bad demonstrations have lowest probability, it would be more sensible to refrain from the region where the expert has high probability (e.g. by maximizing the forward KL). While I admit that the reverse KL can also have the desired effect when $\alpha$ is small enough, I do not think that the proposed optimizaion problem is principled.
2. __Lower bound is loose.__ The method introduces a surrogate loss and justifies it by stating that it is a lower bound of the original objective. However, the fact that one function is dominated by another one is in general not sufficient to justify its use as a surrogate, since the optima can be completely different. The paper states that the bound is tight when the reward implied by the Q-function is uniformly zero, but this doesn't apply to the setting. Hence, the justification of the lower bound optimization is purely empirical.



## Presentation

### Strengths
1. __Well-written and clean figures.__ The presentation is mostly clear and most parts are easy to follow. The figures are neat and provided pseudo-code helpful.
2. __Extensive Appendix.__ The appendix includes along with the derivations, many additional results and ablations.

### Weaknesses
1. __Addionally incorporating entropy regularization is confusing.__ I found the part starting with "To further enhance the efficiency of Q-learning, we adopt the well-known Maximum Entropy (MaxEnt) reinforcement learning framework by incorporating an entropy term into the training objective" confusing. Based on my understanding, Q in Eq.3 (which corresponds to Eq.7 of Sikchi et al. (2024) with $f* = exp(u) - 1$) should already correspond to the soft-Q function as we are optimizing a reverse KL divergence, which induces a maximum entropy RL problem. Indeed, Eq. 4 seems to be essentially the same as Eq. 3, except that we now express the agent's soft-value using $V^\text{soft}$ instead of $Q^\text{soft}$.
2. __Unclear how bad examples are selected.__ The submission states that the negative demonstrates were "selected". Were they sampled randomly, or chosen based on some criteria?


## Significance

### Strengths
1. __Handling negative demonstrations can be sensible in practice.__ The selected problem setting is reasonable. Providing negative examples can be useful for avoiding unsafe behavior or common failure cases.

### Weaknesses

1. __Method appears brittle.__ When using the Q-function instead of the advantage in AWR, the performance drops significantly. This is quite concerning, because it is a minor modification not connected with the theory. ReCOIL obtains good performance when using the advantage function. How would the baseline perform when using $Q$ instead of $A$?

## Originality


### Strength
1. __Novel Approach.__ The paper formulates a novel optimization problem for offline IL with bad demonstrations and derives a practial method to solve it.

### Weaknesses
1. __Does not provide a compelling novel insight.__ I don't think that I learned anything particularly intresting by reading the submission. What would be the main new insight that the paper provides? That avoiding bad demonstrations can help in offline IL? That this can be achieved based on the machinery provided by prior work? That the difference of two KLs is still convex if $\alpha \le 1$?
2. __The contributions are not clearly distinguished from closely related literature.__ For example, the optimization problem is a special case of the problem formulation of Sikchi et al. (cf. Eq. 1), since it corresponds to a KL-regularized RL problem. The little trick of introducing the unlabeled dataset to bring it into this form (Propositon 4.2) was also shown by Sikchi et al. and the resulting objective (Eq. 3/4) can be directly obtained by plugging the reward function $\Psi$ and the convex conjugate into the general form of (Eq.7, Sikchi et al., 2024). Yet, the paper spends a lot a space (full half-column) of presenting these results and does not refer to the more general form. Similarly, Section 5 discusses the use of XQL for optimizing the Q-function, without mentioning that this was already part of ReCOIL.


## Minor Comments
Typos: "an more tractable", "log missing in proof of Proposition 4.1"

---

> ### Author Rebuttal · Authors · 2026-03-31
>
> We thank the reviewer for carefully reading our paper and providing insightful feedback. Please find below our detailed responses to each comment.
>
> ---
>  > The reverse KL is mode-seeking, which can be sensible when minimizing it, but not well-suited for maximization....
>
>  We would like to clarify that DualCOIL "does not optimize the reverse KL" to the bad distribution *in isolation*. The concern about collapse due to maximizing reverse KL alone does not apply in our setting. Importantly, the presence of the expert term $D_{\mathrm{KL}}(d^\pi \| d_G)$ acts as a *strong anchoring mechanism*. The bad-data term only provides a *contrastive correction*, rather than driving the optimization independently.
>
> Your suggestion of using forward KL is interesting. However, in our case, the difference of two forward KL is no longer convex, leading to an intractable learning objective. This would likely require further investigation.
>
> > Lower bound is loose ...
>
> We would like to clarify that our surrogate is *not an arbitrary bound*, but is derived from a first-order approximation of the exponential term, preserving its optimization structure. Importantly, the surrogate offers critical advantages (convex and non-adversarial objective), avoiding instability from the exponential form.
>
> >  Incorporating entropy regularization is confusing ...
>
> We agree that the transition from Eq. (3) to Eq. (4) is somewhat confusing.  To make the derivation more consistent and clearer, we will revise the formulation so that subsequent derivations directly follow from Eq. (3) instead of Eq. (4) (i.e. removing the term $\log \frac{\pi}{\mu^U}$ from the definition of $V^\pi_Q$). This change does not affect the subsequent theoretical results.
>
>  In practical implementation, to incorporate a soft policy and enable implicit $V$-updates, we will then reintroduce the term $\log \frac{\pi}{\mu^U}$ in the definition of $V$. This can facilitate efficient implicit-V updates (as in XQL).
>
> > How bad examples are selected?
>
>  This is just random selection; no criteria is applied.
>
> > When using the Q-function instead of the advantage in AWR, the performance drops significantly.
>
> QWBC works well in our setting. Although it is theoretically equivalent to AWBC, its main advantage is that it *avoids reliance on $V(s)$*, which can be noisy and leads to suboptimal policy learning.
>
>  To further clarify the effect of QWBC, we have applied AWBC to ReCOIL and report the results below. QWBC does not consistently improve performance in ReCOIL, suggesting that *its benefit is specific to our setting*.
> || CHEETAH| ANT| HOPPER| WALKER| PEN| HAMMER| DOOR| RELOCATE|
> |-|-|-|-|-|-|-|-|-|
> | AWBC| 2.0±0.3| 56.2±5.0| **81.0±14.7** | **29.8±14.9** | **79.2±9.6**| **3.4±2.1**| 19.3±7.5| 1.4±1.1|
> | QWBC| **2.2±0.1** | **88.2±11.4**| 78.3±21.6| 3.6±4.7| 24.2±8.5| 0.7±1.2| **59.8±13.2** | **1.6±0.8**|
> > What would be the main new insight that the paper provides?
>
> We would like to clarify that the main contribution of the paper is *not simply that “avoiding bad demonstrations helps”*, but rather in developing a *principled contrastive framework for learning from bad behavior*.
>
> Prior DICE-based methods use KL minimization or weighted sums, while DualCOIL optimizes a difference of KL divergences, explicity enforcing movement away from bad behaviors rather than implicit down-weighting. We believe this is a significant conceptual shift. Another key contribution is in achieving this contrastive distribution matching in a practical way.
>
> > The contributions are not clearly distinguished from closely related literature.
>
>  Sikchi et al. (2024) provides a general and unified framework, but their results are not directly applicable. In particular, our original objective involves a difference of KL divergences, which does not appear anywhere in Sikchi et al. (or other IL work).
>
> We acknowledge that the derivation from Prop. 4.2 to Eq. (3) can follow the framework of Sikchi et al.. This is the *only component* inherited from prior work. There are multiple key contributions of our paper  in the subsequent steps, including (i) the surrogate objective, (ii) non-adversarial training,  (iii) convexity in the Q-space, and (iv) Q-weighted BC policy extraction,
>
> > I would like the authors to relate the lower bound with semi-gradients
>
> Using semi-gradient alone does not eliminate the exponential term. Moreover, our approach is based on an alternative procedure: we fix $V(s')$ and update $Q(s,a)$ using XQL. This can also be viewed as a form of semi-gradient.
>
> ---
> *We hope that the above responses address your questions and concerns. We will revise the paper accordingly. Should you have any further inquiries, we would be happy to consider and respond to them.*

---

> > ### Author Rebuttal · Reviewer_Nta9 · 2026-03-31
> >
> > Thank you for your reply, in particular the additional experiments with Recoil + QWBC. However, my main concerns remain unaddressed.
> >
> > > We would like to clarify that DualCOIL "does not optimize the reverse KL" to the bad distribution in isolation. The concern about collapse due to maximizing reverse KL alone does not apply in our setting. Importantly, the presence of the expert term $D_{\mathrm{KL}}(d^\pi | d_G)$ acts as a strong anchoring mechanism. The bad-data term only provides a contrastive correction, rather than driving the optimization independently.
> >
> > The clarification is unnecessary, as I clearly understood that you are not optimizing this term in isolation. I was trying to make the point that pushing the agent toward the point of lowest bad-expert support, is inherently different from pushing the agent away from regions of high bad-expert support even though (as I have already admitted) it has the desired effect for small $\alpha$, where the good-expert KL-minimization dominates.
> >
> > > We would like to clarify that our surrogate is not an arbitrary bound, but is derived from a first-order approximation of the exponential term, preserving its optimization structure. Importantly, the surrogate offers critical advantages (convex and non-adversarial objective), avoiding instability from the exponential form.
> >
> > Could you please define "preserving its optimization structure"? Are you claiming that it shares the same optima?
> >
> > > We would like to clarify that the main contribution of the paper is not simply that “avoiding bad demonstrations helps”, but rather in developing a principled contrastive framework for learning from bad behavior.
> >
> > > Prior DICE-based methods use KL minimization or weighted sums, while DualCOIL optimizes a difference of KL divergences, explicity enforcing movement away from bad behaviors rather than implicit down-weighting. We believe this is a significant conceptual shift. Another key contribution is in achieving this contrastive distribution matching in a practical way.
> >
> > I understand the contribution of providing a method that performs well in the considered problem setting with bad demonstrations. However, I'm still missing a compelling insight / research finding. Maybe it would help, if you could elaborate on the difference between "enforcing movement away from bad behaviors" and "implicit down-weighting". Doesn't the optimization problem result in lower weights for the weighted BC loss (QWBC)?
> >
> > > Sikchi et al. (2024) provides a general and unified framework, but their results are not directly applicable. In particular, our original objective involves a difference of KL divergences, which does not appear anywhere in Sikchi et al. (or other IL work).
> >
> > A difference of KL divergences is equivalent to the sum (for $\alpha <1$) of a KL and an expected reward/cost, since the entropy terms of both KLs can be grouped (this is precisely Proposition 4.2 in the submission). Hence, as I already stated in my initial review, your objective is a direct instantiation of Eq. 7 in Sikchi et al. (which considers general Bregman divergences). I agree that formulating the problem with a KL maximization term to avoid bad demonstrations is a novel starting point and I also believe that it would be a noteworthy remark that this objective is an instance of the Dual-Q formulation in Sikchi et al. (and can be solved accordingly).
> >
> > > We acknowledge that the derivation from Prop. 4.2 to Eq. (3) can follow the framework of Sikchi et al.. This is the only component inherited from prior work.
> >
> > XQL was also applied by Sikchi et al. which should be cited on top of the original work by Garg et al. to acknowledge that this aspect also follows closely related work.
> >
> > > There are multiple key contributions of our paper in the subsequent steps, including (i) the surrogate objective, (ii) non-adversarial training, (iii) convexity in the Q-space, and (iv) Q-weighted BC policy extraction
> >
> > I would argue that (ii) and (iii) are not key contributions, but properties / motivations to use the surrogate objective.

---

> > > ### Author Response · Authors · 2026-04-03
> > >
> > > We thank the reviewer for their continued engagement in the discussion and for providing insightful feedback. Please find our responses below.
> > >
> > > > Pushing the agent toward the point of lowest bad-expert support, is inherently different from pushing the agent away from regions of high bad-expert support ...
> > >
> > > We agree that maximizing reverse KL in isolation could push the policy toward regions of minimal bad density, which is not desirable. However, our objective is a *joint contrastive objective*  where the expert term provides a strong anchor, preventing collapse to arbitrary low-density regions.
> > >
> > > Our formulation can be interpreted as optimizing a **density difference**: $\log {d_G}-\alpha \log {d_B},$ which encourages the policy to prefer regions where *good behavior is relatively more likely than bad*, rather than simply seeking regions where bad density is minimal. To illustrate our point, we can give a simple MDP with four state-action pairs:
> > > | $(s,a)$ | $d_G$ | $d_B$ |
> > > |--------|------|------|
> > > | A      | 0.4  | 0.01 |
> > > | B      | 0.35 | 0.3  |
> > > | D      | 0.05 | 0.5  |
> > > | C      | 0.01 | 0.01 |
> > >
> > > With $\alpha = 0.5$, the reward in our learning objective (in simple form) can be computed as:$r(s,a) = \log d_G(s,a) - 0.5 \log d_B(s,a)$, giving: $r(A) \approx 1.38$, $r(B) \approx -0.45$, $r(C) \approx -2.30$ ,$r(D) \approx -2.65$
> > >
> > > Thus, the learning policy will prefer **A**, which has high good density and very low bad density.  In contrast:
> > >   - B is less preferred due to higher bad density
> > >   - C is not preferred despite low bad density (since good is also very low)
> > >   - D is strongly penalized due to high bad density
> > >
> > > This example demonstrates that our objective does not simply seek regions with minimal bad density, but instead favors regions where good behavior is sufficiently strong relative to bad behavior, avoiding both high-bad regions and low-information regions.
> > >
> > > > Could you please define "preserving its optimization structure"? Are you claiming that it shares the same optima?
> > >
> > > Thank you for the question. We do not claim that the surrogate and the original objective share the same optimum. Indeed, this is not theoretically true, as we approximate an exponential function with a linear one.
> > >
> > > Our key point is that the linear approximation preserves the *optimization behavior*: increasing or decreasing $x$ leads to consistent changes in both $\exp(x)$ and its approximation $(x + 1)$. Thus, the surrogate maintains aligned optimization directions.
> > >
> > > We acknowledge that this introduces an approximation gap. However, this is a deliberate trade-off: we accept some approximation error in exchange for a *more tractable and stable objective*. In practice, we found that directly optimizing the exponential form is highly unstable, which motivates our linear approximation.
> > >
> > > > ... elaborate on the difference between "enforcing movement away from bad behaviors" and "implicit down-weighting" ..
> > >
> > >  Our point here is that we explicitly incorporate the term $D_{\mathrm{KL}}(d^{\pi} \| d^B)$, which enforces *movement away from bad behaviors* in a direct and principled manner.
> > >
> > > In contrast, SafeDICE — one of the prior works aiming to handle bad behavior—formulates the problem as a KL minimization with a negatively weighted mixture:$\min_{d^\pi} D_{\mathrm{KL}}\left(d^{\pi} \,\middle\|\, \frac{d^U - \alpha d^B}{1 - \alpha}\right).$ This formulation can be problematic. For example, when the unlabeled data $d^U$ is similar to $d^B$, the objective effectively reduces to: $\min_{d^\pi} D_{\mathrm{KL}}(d^{\pi} \| d^B),$ which would incorrectly push the policy **toward bad behavior**.
> > >
> > > > Your objective is a direct instantiation of Eq. 7 in Sikchi et al. (which considers general Bregman divergences)
> > >
> > > Thank you for the comment. We agree that Eq. (3) can be solved using techniques from Sikchi et al., and it is indeed worth noting that, after Prop. 4.2, our learning problem can be cast within their general framework.
> > >
> > > Our work was developed based on the observation that directly optimizing Eq. (3) does not yield good performance in practice (as shown in our ablation study), which motivates our surrogate objective that leads to significantly more stable and robust training.
> > >
> > > Therefore, while our objective can be viewed as an instance of Sikchi et al. (after our reformulation in Prop 4.2), their techniques alone is not sufficient to achieve stable learning in our setting.
> > >
> > > > XQL was also applied by Sikchi et al. ..
> > >
> > >  We will revise the paper to include appropriate citations, acknowledging that XQL was also applied by Sikchi et al., in addition to the original work by Garg et al.
> > >
> > >  > I would argue that (ii) and (iii) are not key contributions ...
> > >
> > > We agree with this point. The primary contribution is the *surrogate objective* itself, while (ii) and (iii) are better interpreted as properties  that justify its use.
> > >
> > > ---
> > >
> > > We hope the above responses further clarify our motivation and contributions. We will revise the paper accordingly.

---

### Official Review · Reviewer_f35R · 2026-03-13

**Soundness:** 3
**Presentation:** 3
**Significance:** 2
**Originality:** 2
**Overall Recommendation:** 4
**Confidence:** 4

**Summary:**

This paper proposes **DualCOIL**, an offline imitation learning method designed to learn from demonstrations of mixed quality. The approach frames the learning problem as a hybrid KL objective, minimizing the distance to desirable behaviors while maximizing the distance to undesirable ones. The method also integrates existing techniques, such as XQL and weighted behavior cloning, to improve optimization stability. The authors evaluate DualCOIL on several state-based robotics benchmarks, demonstrating strong performance compared to baselines, including SafeDICE and DWBC.

**Compliance With Llm Reviewing Policy:**

Affirmed.

**Key Questions For Authors:**

- Can the authors provide a conceptual comparison table summarizing different DICE-style designs, highlighting which design choices contribute most to stable offline RL?

- Given the extensive literature on DICE since 2021, it would be helpful to contextualize DualCOIL more clearly in terms of its novel contribution versus existing techniques.

**Limitations:**

Yes

**Strengths And Weaknesses:**

## Strengths
- **Clear presentation:** The derivations and algorithmic details are explained in a straightforward and understandable way.
- **Thorough evaluation:** The experimental comparisons are comprehensive, and the proposed method achieves strong empirical performance.
- **Potential Practical impact:** The method addresses the important challenge of learning from mixed-quality demonstrations, which is relevant for real-world offline imitation learning.

---

## Weaknesses
- **Limited novelty:** The hybrid KL learning objective is not fundamentally new; prior work such as SafeDICE has explored similar approaches using mixed KL terms. DualCOIL appears largely as a combination of existing techniques to stabilize DICE-style optimization, without a clear conceptual innovation. It is unclear what uniquely differentiates this method from prior DICE-based approaches.

- **Sensitivity to hyperparameters:** The method appears sensitive to the choice of alpha and beta, which may affect stability and generalization.

- **Conceptual context:** While the empirical evaluation is thorough, the work could benefit from more discussion of related DICE-style methods, particularly regarding how design choices influence stability and performance.

---

> ### Author Rebuttal · Authors · 2026-03-30
>
> We thank the reviewer for carefully reading our paper and providing insightful feedback. Please find below our detailed responses to each comment.
>
> ---
> > Limited novelty: The hybrid KL learning objective is not...
>
> Thank you for the comment. We believe our method provides several fundamental novelties that clearly distinguish it from prior DICE-based and imitation learning methods.
>
> Unlike prior DICE methods (e.g., SafeDICE) that use KL minimization or weighted sums, DualCOIL optimizes a difference of KL divergences, explicitly enforcing repulsion from bad behaviors rather than implicit down-weighting.
>
> We prove this difference-of-convex objective becomes convex (under reasonable conditions), enabling tractable optimization—this property is absent in prior DICE formulations.
>
> SafeDICE and related methods remain within a single-divergence minimization paradigm, whereas DualCOIL introduces a contrastive distribution-matching framework (imitate good, avoid bad) derived from first principles.
>
> The new formulation leads to consistent empirical gains (Table 1, Fig. 1) and uniquely leverages bad data, which prior methods fail to exploit.
>
> > Sensitivity to hyperparameters: The method appears sensitive...
>
> We agree that $\alpha$ can be sensitive for some tasks (e.g., Hammer), as it balances the influence of good and bad demonstrations.
>
> However, our experiments show a **robust range of $\alpha$ values** (e.g., $\alpha < 0.4$) where performance remains consistently strong, providing practical guidance for selecting $\alpha$.
>
> Similarly, sensitivity to $\beta$ is consistent with prior work on weighted behavior cloning. We observe stable performance across a **range of sufficiently large $\beta$ values** (e.g., $\beta > 15$), again providing useful guidance.
>
> Overall, while some sensitivity to key hyperparameters is unavoidable in offline RL, our results show that DualCOIL admits **broad, stable regimes** for both $\alpha$ and $\beta$.
>
> > Can the authors provide a conceptual comparison table summarizing different DICE-style designs...
>
> > Conceptual context: While the empirical evaluation is thorough, the work could benefit from more discussion of related DICE-style methods...
>
> Thank you for the suggestion. We agree that a conceptual comparison is valuable. Below we summarize key design differences among DICE-style methods and highlight factors contributing to stable offline RL:
> | Method| Objective Type| Handles Bad Data Explicitly|Stability Mechanism|
> |-|-|-|-|
> |ValueDICE|KL minimization| No| Saddle-point optimization, adversarial trainning|
> | SMODICE, DemoDICE| KL minimization (with mix)| No| DICE-based standard  reformulation, non-adversarial|
> | SafeDICE| KL minimization + weighting mechanism  | Partial (sensitive to unlabeled data quality, due to the KL minimization approach) | DICE-based standard  reformulation, non-adversarial|
> | DualCOIL (Ours)  | KL difference (contrastive)| Yes (explicit repulsion)| DC structure + convex reformulation + convex non-adversarial surrogate|
>
> Key takeaway: Prior methods focus on **matching expert distributions**, while DualCOIL introduces a **contrastive distribution-matching framework (match good, mismatch bad)** with a provably tractable objective. We believe this explicit contrastive design, together with convex reformulation, is a key factor behind improved stability and performance. We will incorporate this comparison into the revised paper.
>
> > Given the extensive literature on DICE since 2021, it would be helpful to contextualize DualCOIL more clearly in terms of its novel contribution versus existing techniques.
>
> Thank you for the suggestion. We agree that clearer positioning is important. While DualCOIL builds on the DICE framework, its novelty lies in **extending distribution-matching IL beyond pure divergence minimization**.
>
> Existing DICE-based methods (since 2021) primarily focus on:
>   - **matching expert distributions** via KL or f-divergence minimization,
>   - optionally incorporating unlabeled or suboptimal data through **reweighting or mixture modeling**,
>   - but **do not explicitly model avoidance of undesirable behaviors** at the objective level.
>
> In contrast, DualCOIL introduces:
>   1. A **difference-of-KL objective** that explicitly captures *“imitate good, avoid bad”*
>   2. A **non-trivial convexity result** for this DC objective (when $\alpha \le 1$
>   3. A **principled dual RL formulation** derived from this objective (rather than heuristic reward shaping)
>
> Therefore, DualCOIL is not a direct extension of existing DICE methods, but a **conceptual generalization from single-distribution matching to contrastive distribution matching**.
>
> We will revise the paper to better emphasize this distinction and clarify its relationship to prior DICE-based approaches.
>
> ---
> *We hope that the above responses address your questions and concerns. We will revise the paper accordingly. Should you have any further inquiries, we would be happy to consider and respond to them.*

---

> > ### Author Rebuttal · Reviewer_f35R · 2026-04-06
> >
> > Thank you for the update!

---

### Official Review · Reviewer_S26o · 2026-03-13

**Soundness:** 3
**Presentation:** 3
**Significance:** 3
**Originality:** 3
**Overall Recommendation:** 4
**Confidence:** 1

**Summary:**

This paper introduces DualCOIL, a novel offline imitation learning framework that learns from three types of datasets: good (expert) demonstrations, bad (undesirable) demonstrations, and a large unlabeled dataset. The core idea is to formulate the learning objective as the difference of two KL divergences: minimizing divergence from the good data while maximizing divergence from the bad data. To improve numerical stability, they propose a lower-bound surrogate objective and introduce a Q-weighted Behavior Cloning (QW-BC) method for policy extraction.

**Compliance With Llm Reviewing Policy:**

Affirmed.

**Final Justification:**

This paper presents a novel and practically motivated offline imitation learning framework that effectively leverages good, bad, and unlabeled data, and the authors’ rebuttal including additional experiments and clarifications on noisy or similar demonstrations and comparisons to reward-shaping baselines fully addresses my concerns, reinforcing my positive assessment and original weak accept (4) recommendation.

**Key Questions For Authors:**

- What happens when good and bad behaviors are similar? In tasks where the difference between expert and suboptimal behavior is subtle (e.g., slight timing differences), does the KL divergence formulation still provide a clear separation?
- How does DualCOIL perform if the "bad" dataset is noisy? For instance, what happens if it is a mixture of truly bad trajectories and some expert-level trajectories mistakenly labeled as bad?
- How does DualCOIL compare to a simpler baseline where one simply assigns a negative reward to transitions in the "bad" dataset and uses a standard offline RL algorithm? Does the explicit KL divergence manipulation offer significant advantages over simple reward shaping?

**Limitations:**

yes

**Strengths And Weaknesses:**

# Strengths：
The paper tackles an underexplored but highly practical setting: learning from explicitly labeled good and bad demonstrations alongside unlabeled data. This is relevant for real-world applications where both positive and negative examples are available.
# Weaknesses：
The method relies on accurately estimating density ratios via discriminators. If the good or bad datasets are too small or unrepresentative, these estimates may be noisy, potentially degrading performance.

---

> ### Author Rebuttal · Authors · 2026-03-30
>
> We thank the reviewer for carefully reading our paper and providing insightful feedback. Please find below our detailed responses to each comment.
>
> ---
>
> > The method relies on accurately estimating density ratios via discriminators. If the good or bad datasets are too small or unrepresentative, these estimates may be noisy, potentially degrading performance.
>
> Thank you for the comment. We agree that if the amount of high-quality (good) and low-quality (bad) data is limited or noisy, the performance can deteriorate. This is, however, a common phenomenon in IL (and broadly in machine learning), as most algorithms require sufficiently well-labeled good (or bad) demonstrations to learn effectively; otherwise, they tend to produce low-quality policies.
>
> In our work, we require only a small amount of high-quality (good) demonstrations (typically expensive to collect), while allowing for a larger amount of low-quality (bad) demonstrations (typically easier to collect) , which is a  practical and realistic setting. We achieved better performance compared to SOTA on dataset of similar quality.
>
> > What happens when good and bad behaviors are similar? In tasks where the difference between expert and suboptimal behavior is subtle (e.g., slight timing differences), does the KL divergence formulation still provide a clear separation?
>
> Thank you for the comment. When good and bad behaviors are similar, our formulation remains robust and stable. In Eq. (1), if $d^G \approx d^B$, the objective can be approximated as $(1 - \alpha)\, D_{\mathrm{KL}}(d^\pi \| d^G)$. Under the assumption $\alpha < 1$, this reduces to a standard IL objective, encouraging the policy to mimic expert behavior. Thus, our method does not degrade and still learns effectively.
>
> Empirically, we observe that our method remains effective even when bad data comes from medium-quality policies (i.e., not drastically different from expert), suggesting that clear separation between good and bad data is not required for performance gains.
>
> > How does DualCOIL perform if the "bad" dataset is noisy? For instance, what happens if it is a mixture of truly bad trajectories and some expert-level trajectories mistakenly labeled as bad?
>
> Thank you for the comment. Our method is already evaluated in a partially noisy setting in Table 1 of the manuscript, where the bad dataset is not purely random failure but consists of medium-performance or cloned trajectories. The strong performance in that setting suggests that DualCOIL is robust to moderate noise in the bad dataset.
>
> To further address your comment, we conducted additional experiments by injecting expert demonstrations into the bad dataset (i.e., introducing mislabeled data). Specifically, we keep the good dataset $\mathcal{B}^G$ to a single expert trajectory, while the bad dataset $\mathcal{B}^B$ contains 10 random trajectories (for CHEETAH and HOPPER) and 25 cloned trajectories (for HAMMER and RELOCATE), following the same setup as Table 1 in the manuscript.
>
> As can be expected, adding expert trajectories to bad dataset degrades the performance due to conflicting signals. However, as can be seen below, the performs degrades gracefully until 10 expert trajectories are added to the bad dataset, which is when there is equal representation of bad and good in bad dataset.
>
> | Number of Expert in $\mathcal{B}^B$ |CHEETAH|HOPPER|HAMMER|RELOCATE|
> |-|-|-|-|-|
> |0|86.7±2.2|93.6±9.2|74.3±8.0|92.1±5.0|
> |1|86.5±0.2|62.8±11.8|60.6±3.4|49.7±4.3|
> |3|84.9±1.8|36.5±45.4|55.5±4.3|10.5±2.1|
> |5|84.9±1.9|2.6±2.3|54.8±5.7|0.6±0.4|
> |10|78.0±3.1|3.7±4.1|48.4±6.9|0.5±0.3|
> |25|2.2±0.1|1.7±0.0|3.7±2.3|0.1±0.1|
>
> > How does DualCOIL compare to a simpler baseline where one simply assigns a negative reward to transitions in the "bad" dataset and uses a standard offline RL algorithm? Does the explicit KL divergence manipulation offer significant advantages over simple reward shaping?
>
> Thank you for the comment. We already have an ablation study comparing our method with the simple reward shaping baseline (assigning +1 to expert data and −1 to bad data, followed by standard offline RL). The results (see D.8 in the appendix) consistently show that this reward shaping approach performs worse than DualCOIL.
>
> In general, simple reward shaping tends to **collapse the information** from good and bad demonstrations into scalar signals. In contrast, DualCOIL formulates learning as a **distribution matching and dismatching problem**, explicitly encouraging the policy to move toward good behavior while avoiding bad behavior. This leads to a more principled objective and ensures **consistency between reward learning and policy optimization**, whereas naive reward shaping can introduce bias or unintended consequences.
>
> ---
>
> *We hope that the above responses address your questions and concerns. Should you have any further inquiries, we would be happy to consider and respond to them.*

---

> > ### Author Rebuttal · Reviewer_S26o · 2026-04-03
> >
> > Thank you for the detailed rebuttal and additional experiments. The explanations and new results clarify how DualCOIL behaves under limited, noisy, or similar good/bad data, and the comparisons with reward-shaping baselines further support the advantages of your approach.
> >
> > My assessment of the remaining weaknesses—dependence on discriminator estimates and data quality—remains, but these are common to similar baselines. Overall, the rebuttal increases clarity and confidence in the method, and I maintain my positive assessment and original score.

---

### Decision · Program_Chairs · 2026-04-30

**Decision:**

Accept (regular)

**Comment:**

This paper presents DualCOIL, an offline imitation learning framework that leverages expert, undesirable, and unlabeled demonstrations by optimizing a difference of KL divergences, and derives a convex non-adversarial surrogate with a Q-weighted BC policy extraction. The rebuttal resolved most concerns from three reviewers by adding experiments on mislabeled bad data, reward-shaping baselines, varying the number of demonstrations, and hyperparameter sensitivity. One reviewer raised substantive concerns that remained only partially addressed: after Proposition 4.2, the objective becomes a direct instance of the Dual-Q framework of Sikchi et al. (2024), which limits the conceptual novelty beyond introducing the KL-maximization term. Also, the linear lower-bound surrogate of the exponential term is a first-order heuristic whose optima provably differ from the original DC objective, so its justification remains empirical rather than principled. These are legitimate limitations, and prior-work attribution should be strengthened accordingly. Nonetheless, the contrastive formulation, the convexity result under $\alpha \le 1$, and consistent empirical gains, constitute a solid and practically meaningful contribution.